# Lived experiences of frontline healthcare providers offering maternal and newborn services amidst the novel corona virus disease 19 pandemic in Uganda: A qualitative study

Herbert Kayiga[1]*, Diane Achanda Genevive[2], Pauline Mary Amuge[3], Andrew Sentoogo Ssemata[4], Racheal Samantha Nanzira[2], Annettee Nakimuli[1]

1 Department of Obstetrics and Gynecology, Makerere University College of Health Sciences, Kampala, Uganda, 2 Kawempe National Referral Hospital, Kampala, Uganda, 3 BAYLOR College of Medicine, Kampala, Uganda, 4 Department of Psychiatry, Makerere University College of Health Sciences, Kampala, Uganda

* hkayiga@gmail.com

## Abstract

### Background

The COVID-19 pandemic has brought many health systems in low resource settings to their knees. The pandemic has had crippling effects on the already strained health systems in provision of maternal and newborn healthcare. With the travel restrictions, social distancing associated with the containment of theCOVID-19 pandemic, healthcare providers could be faced with challenges of accessing their work stations, and risked burnout as they offered maternal and newborn services. This study sought to understand the experiences and perceptions of healthcare providers at the frontline during the first phase of the lockdown as they offered maternal and newborn health care services in both public and private health facilities in Uganda with the aim of streamlining patient care in face of the current COVID-19 pandemic and in future disasters.

### Methods

Between June 2020 and December 2020, 25 in-depth interviews were conducted among healthcare providers of different cadres in eight Public, Private-Not-for Profit and Private Health facilities in Kampala, Uganda. The interview guide primarily explored the lived experiences of healthcare providers as they offered maternal and newborn healthcare services during the COVID-19 pandemic. All of the in depth interviews were audio recorded and transcribed verbatim. Themes and subthemes were identified using both inductive thematic and phenomenological approaches.

### Results

The content analysis of the in depth interviews revealed that the facilitators of maternal and newborn care service delivery among the healthcare providers during the COVID-19

**Data Availability Statement:** All relevant and anonymized data are within the paper and its Supporting information files.

**Funding:** This project was supported by The Government of Uganda through the Makerere University Research Innovations Fund (RIF) project of Makerere University. The content is solely the responsibility of the authors and does not necessarily represent the official views of the Government of Uganda. The funders had no role in study design, data collection and analysis, decision to publish, or preparation of the manuscript.

**Competing interests:** The authors have declared that no competing interests exist.

**Abbreviations:** ANC, Antenatal Clinic Attendance; **CME**, Continuous Medical Education; **COVID-19**, Novel Corona virus Disease; **HIV**, Human Immune Virus; **ICU**, Intensive Care Unit; **MERS-CoV**, Middle Eastern Respiratory Syndrome Corona virus; **MMR**, Maternal mortality rate; **PPE**, Personal Protective Equipment; **RDC**, Resident District Commissioners; **RT-PCR**, Reverse Transcription Polymerase Chain Reaction; **SARS-CoV-2 virus**, Severe Acute Respiratory Syndrome Corona virus-2; **UN**, United Nations; **WHO**, World Health Organization.

pandemic included; salary bonuses, the passion to serve their patients, availability of accommodation during the pandemic, transportation to and from the health facilities by the health facilities, teamwork, fear of losing their jobs and fear of litigation if something went wrong with the mothers or their babies. The barriers to their service delivery included; lack of transport means to access their work stations, fear of contracting COVID-19 and transmitting it to their family members, salary cuts, loss of jobs especially in the private health facilities, closure of the non-essential services to combat high patient numbers, inadequate supply of Personal Protective equipment (PPE), being put in isolation or quarantine for two weeks which meant no earning, brutality from the security personnel during curfew hours and burnout from long hours of work and high patient turnovers.

## Conclusion

The COVID-19 Pandemic has led to a decline in quality of maternal and newborn service delivery by the healthcare providers as evidenced by shorter consultation time and failure to keep appointments to attend to patients. Challenges with transport, fears of losing jobs and fear of contracting COVID-19 with the limited access to personal protective equipment affected majority of the participants. The healthcare providers in Uganda despite the limitations imposed by the COVID-19 pandemic are driven by the inherent passion to serve their patients. Availability of accommodation and transport at the health facilities, provision of PPE, bonuses and inter professional teamwork are critical motivators that needed to be tapped to drive teams during the current and future pandemics.

## Introduction

On March 11th, 2020, the World Health Organisation (WHO) classified the novel Coronavirus disease (COVID-19), caused by the Severe Acute Respiratory Syndrome Corona virus-2 (SARS-CoV-2 virus), as a global pandemic and emergency [1, 2]. As of 10th August 2021, globally there were 202.1 million confirmed cases of COVID-19 and 4.29 million deaths [3]. In Africa as of 10th August 2021, there were 5.14 million confirmed cases and 122,025 deaths from COVID-19, which is lower than the 78.6 million confirmed cases and 2.03 million deaths in Americans [3] and 61.2 million confirmed cases in Europe with 1.23 million deaths from COVID-19 [3].

Uganda reported her first COVID-19 case on the 21st March 2020 [4]. Since then the number of confirmed cases had reached 95,723 as of 06th August 2021 with 2,783 deaths reported by the Uganda Ministry of Health [5]. The COVID-19 pandemic took Uganda by surprise [6, 7]. With 1.4 million HIV positive patients, 800,000 diabetic patients and 100,000 TB positive sputum patients, the Ugandan health system was already overstretched [8].

Healthcare in Uganda is offered mainly by public (70%), private-Not for profit (20%) and private health facilities (10%) [9, 10]. Public health facilities are structured in the following categories; National and Regional referral hospitals, general hospitals, district hospitals/ Health centre IVs (offering care to a population of 100,000 both in and outpatient services and emergency surgeries), Health Centre III (serving a population of 20,000 at the sub county level offering mainly outpatient and maternity services), Health centre II (serving a population of 5,000 and being run by an enrolled midwife) and the Health centre I (linking the community to the health system and being run by the village health teams with or without formal training).

Care in the public facilities is free [11]. Private-Not-for Profit health facilities are mainly faith based facilities that offer care at a subsidized cost. The private health facilities are run by individuals or institutions with no exact control on how care is billed [12].

With lessons from other low and middle income settings, like Vietnam where lockdowns, extensive contact tracing and social distancing had resulted in barely any mortality attributed to COVID-19 [13, 14], and more case fatality rates from COVID-19 in the United States [15] and Europe [16] where preventative measures were not fully implemented, Uganda instigated a nationwide lockdown to contain the COVID-19 pandemic.

With no clear cure to COVID-19 [17, 18], like other African countries (South Africa, Malawi, South Sudan, Kenya, Ghana, Nigeria, and Rwanda) [19], Uganda took a number of measures to contain the COVID-19 pandemic. Public gatherings, shopping malls, public domestic and international travels were closed from the 18th March 2020. This was after recommendations of self-quarantine declared from 10th March 2020 for all travellers for two weeks were not adequate to contain the COVID-19 threat in the country. The government closed all the Ugandan borders on 23rd March 2020. With sprouting COVID-19 cases, the authorities suspended all public transport on 25th March 2020 [7]. This was later followed by a nationwide lockdown and night curfews for the first time in Uganda for two weeks (from 1st April 2020) [7]. Before this, there had only been regional lockdowns like in the early 2000s to contain the Ebola outbreaks and civil wars seen around 1980 to 1985. All outdoor exercises were banned on 8th April 2020. After the two weeks, the Ugandan authorities extended the lockdown on the 14th April 2020 up to 5th May 2020. Though eased a bit with reduction on the travel restrictions, the lockdown was extended for another two weeks. The lockdown was finally eased on 4th June 2020 but the curfew measures were left in place.

The healthcare system in Uganda like other low and middle countries faced challenges such as high patient load amidst limited human resource, infrastructure and frequent stock-outs of equipment, drugs and supplies even before the COVID-19 pandemic [8, 19, 20]. Prior to the COVID-19, Uganda through strategies like five year Health Sector Strategic Plans for the past two decades had reduced maternal mortality rates from 500 in 2000 to 375 deaths per 100,000 live births [21, 22]. The four visit antenatal attendance (ANC) was at 59.9% from 33.1% in 2011 [23]. The unmet need for modern contraception had reduced to 26% from 30% in 2016 [22, 24]. The fertility rate in Uganda stands at 4.3 currently from 5.3 in 2000 [21]. The postnatal care was still below optimal levels in Uganda at 54.3% [22]. The neonatal mortality was at 19.9 per 1000 live births before COVID-19 from 33 deaths per 1000 live births [21]. Neonatal tetanus protection had reached 85% as compared 52% in 2000 [25]. Marked progress had also been seen in the BCG immunization at 1 year with 88% while that of Haemophilus influenzae type B (Hib) and Diphtheria, Pertussis (whooping cough), and Tetanus (DPT) vaccine coverage was at 93% [22] before COVID-19 pandemic.

COVID-19 has exerted enormous pressure on National Health Service programs in many African countries like Expanded Program on Immunization [26] as result of closure of some of the vaccination clinics with some of the healthcare providers put in quarantine when suspected or confirmed with COVID-19 or shifted to manage COVID-19 patients [19].

Despite evidence of routine childhood immunization benefit over COVID-19 associated risks with the vaccination clinics [27], the Ministry of Health of Uganda has already reported a decline in the current immunization coverage during the COVID-19 pandemic [28]. Similar trends in immunization coverage have also been reported in South Sudan, Zimbabwe, South Africa and Nigeria [19].

In Uganda, there are 4,600 deliveries per day [29]. There's evidence that skilled birth attendance can reduce preventable maternal and newborn death [20]. Interruption in access to quality maternal and newborn health services with the travel restrictions in place to curb the

COVID-19 could put over 10,000 lives of both women and their babies in danger every single day of the COVID-19 pandemic [6].

As of 7[th] July 2021, 37 Ugandan health workers had died of COVID-19 [30]. The Mulago National Referral Hospital COVID-19 management Unit had only eight Intensive Care Unit (ICU) beds by then and also with reported shortage in supply of oxygen. This low capacity made even the healthcare providers infected with COVID-19 fail to access this critical care when needed [31]. Despite the low human resource available for maternal and newborn health, some healthcare providers were deployed to manage the COVID-19 patients [6]. This could have appreciably affected maternal and newborn healthcare delivery in Uganda as there were fewer frontline healthcare providers on ground to care for the mothers and their babies.

It's against this background that this study sought to understand the lived experiences and perceptions of the healthcare providers offering maternal and newborn services during the first phase of the lockdown to contain the COVID-19 pandemic in Uganda with the aim of streamlining patient care in the current and similar future disasters.

## Materials and methods

### Study design

We conducted this embedded qualitative study as part of a bigger study that assessed the impact of COVID-19 pandemic on the provision of Maternal and Newborn healthcare services in eight health facilities in Kampala, Uganda between June 2020 and December 2020 [32] during the first phase of the lockdown. We used the phenomenological [33, 34] and inductive thematic approaches [35] to explore the lived experiences and perspectives of 25 healthcare providers as they offered maternal and newborn services in the eight selected facilities in Kampala using in depth interviews.

### Study setting

This study was conducted in eight health facilities (two Private hospitals, three Private-Not-for Profit hospitals and three Public health facilities) in Kampala, Uganda. These eight facilities were purposively selected because they are the biggest service providers in the three sectors (public, private-not-for profit and private) offering maternal and newborn health care in Kampala. All of the eight health facilities had most of the different cadres of healthcare providers for maternal and newborn health with brief description provided in Table 1.

### Participant recruitment and sampling

Prior to participant recruitment, we sought permission from the different hospital institutional review boards. After obtaining permission, we met the different hospital administrators who later allowed us to meet the healthcare providers in maternal and newborn health based on their availability and convenience. We purposed to meet healthcare providers of different cadres offering maternal and newborn health services. These included obstetricians/gynecologists, theatre in-charges, nurse midwives, medical doctors, ward in-charges, nurse in-charges of immunization, antenatal, postnatal and family planning clinics. The selected healthcare providers were then given two contacts of the Principal investigator and the research team. We purposively interviewed 25 healthcare providers at the eight selected health facilities using in depth interviews which were preferred to focus group discussions to minimize any spread of the pandemic. Disinfection protocols were observed prior to the interviews. All the in depth interviews were administered in English, the official language used in Uganda in quiet rooms at the different selected health facilities as recommended by the hospital administrators.

**Table 1. Characteristics of the eight health facilities offering maternal and newborn health services in Kampala, Uganda.**

| Hospital | Nature of Health facility | Level of Care | No. of beds and population served | No. healthcare providers | Duration of work | No. of deliveries/ year | Cost of service delivery | Service delivered |
|---|---|---|---|---|---|---|---|---|
| 1. | Public | National Referral hospital | Serves a population of 4.5 million, with a bed capacity of 900 | 500 | 24 hours a day, 7 days a week. | 24,526 | Free | Teaching hospital. Offers free Maternal and Newborn services |
| 2. | Public | Regional referral hospital | Serves a population of 3 million, Bed capacity of 100. | 356 | 24 hours a day, 7 days a week. | 15,000 | Free | Teaching hospital Offers free Maternal and Newborn services |
| 3. | Public | Health Centre III | Serves a population of 200,000. Bed capacity of 30. | 28 | 8 hours, 5 days a week | 5,336 | Free | Maternity and newborn health services, OPD services |
| 4. | Private-Not-for-Profit health facility | Hospital | 361 beds. | 300 | 24 hours a day, 7 days a week | 5,500 | Subsidized cost | Offers both Outpatient and in-patient care It is involved in patient care. Research and teaching. internship site for medical graduates |
| 5. | Private-Not-for-Profit health facility | Hospital | 274 beds | 350 | 24 hours a day, 7 days a week. | 6,832 | Subsidized care | Offers maternal and newborn care services. Internship site. The hospital also offers specialized inpatient and outpatient services |
| 6. | Private-Not-for-Profit health facility. | Hospital | 350 beds. | 347 | 24 hours a day, 7 days a week. | 5,000 | Subsidized care | This hospital offers most of the specialist services in maternal and newborn health. It is also an internship site for medical graduates and a training site for clinical officers, nurses and radiology students. |
| 7. | Private | Hospital | 60 beds. | 60 | 24 hours a day, 7 days a week. | 1,000 | Cash or privately insured care | It offers specialist services to both privately insured and cash patients. Offers all maternal and newborn health services. |
| 8. | Private | Hospital | 80 beds. | 45 | 24 hours a day, 7 days a week. | 1,100 | Cash or privately insured care | The hospital has been providing primary, secondary and some tertiary health care services for the past 25 years. It offers all specialist service |

**Inclusion criteria.** Healthcare providers actively involved in maternal and newborn health service delivery at any of the eight selected health facilities during the study period that consented to participate in the study were recruited.

**Exclusion criteria.** Healthcare providers involved in maternal and newborn health services at the eight selected health facilities who were on leave or inaccessible physically during the study period were excluded.

**Staff training and recruitment.** We had three teams on the study. Team 1 was in charge of data collection. The team was composed of two researchers and two field note takers. The two researchers had doctoral degrees and were familiar with the local hospital settings. This team had research training for three days. They were trained on how to identify and interview potential participants. They were also trained on participant recruitment while observing the research ethics in accordance to the Declaration of Helsinki [36]. The two field note takers were fluent in English and Luganda, the locally spoken language. Team 2 was in charge of data analysis. It was composed of Principal investigator and one administrator. This team had to ensure transcription accuracy and data analysis. Team 3 was composed of two independent researchers whose task was checking rigor according to the Lincoln—Guba criteria [37].

**Data collection.** After obtaining informed consent from the participants, 25 healthcare providers had in depth interviews by two doctoral degree level interviewers between June 2020

and December 2020. All interviews were administered in English. Two note takers captured the participants' non-verbal expressions with their consent in addition to the field notes. After ascertaining data saturation with no new emerging themes, we stopped the data collection [38, 39]. The interviews lasted between 45 to 90 minutes. The interviews captured the participant socio-demographic information, the way they perceived service delivery before and during the COVID-19 pandemic, the facilitators and barriers to quality maternal and newborn service delivery and their recommendations to optimal service delivery in future disasters or pandemics using open ended questions. Whenever clarity was needed, more specific questions were raised by the interviewers so that all the required information was collected. All of the interviews were tape recorded.

**Quality control.** Two interviewers and two note takers were trained prior to the data collection. A pilot study was carried out with four healthcare providers to pretest and modify the interview guide. Data from the pilot study was also included in the analysis as the healthcare providers in the pilot were not included in the main study. The interviews were tape recorded and transcribed verbatim immediately after the interviews. The transcriptions were compared with field notes throughout the study period. We ensured that the coordinators of the interviews or discussions didn't participate in the analysis but critiqued the results from the analysis and ensured that these results conformed to their expectations from the discussions. This was done to validate the study findings and also ensure quality in the study. Field notes and transcripts, codes and their interpretations were made by separate teams of investigators. Data was backed up on hard drives, online databases and two computers. The research materials were kept under restricted access by only authorized staff for participant confidentiality and privacy.

**Ethical consideration.** We obtained ethical approvals from The AIDS Support Organisation (TASO) Institutional Ethics review board, (TASOREC/064/2020-UG-REC-009), and Uganda National Council of Science and Technology (HS924ES). We also obtained administrative clearances from the eight health facilities. Verbal and written informed consents were obtained from all study participants after an elaborate explanation of the study.

Participants were reimbursed for participating in the study in form of transport refunds. Participants were reassured that participating in the study was voluntary and that they could opt out of the study without compromising the relationship with the research team. Confidentiality and participants rights were observed throughout the study. All participants' data (audiotapes, records, transcripts and notes) were kept in a secure location accessible only to study personnel. Study participants were identified by pseudonyms rather than actual names in the final report.

## Data analysis

Using the inductive thematic approach [35] and Colaizzi's process of data analysis for phenomenological studies [34, 39], the research team took the following steps: data was prepared by typing out the interviews, thereafter using sentences, phrases or paragraphs, generated meaning units from the context of the participants' voices. We then converted the concepts generated into codes (text coding) using semantic tags. The primary codes were then generated and meaning units shortened to formulate 'compressed meaning units'. We later revised the text codes comparing similarities and differences between the codes thereby integrating what appeared as similar codes. We then critically looked at all the transcript steps and codes and classified them based on their relationships or differences. We ensured reliability of the codes, and then revised the classes. Data was coded and analyzed manually using a framework matrix developed using an Excel workbook built after a detailed and careful process of

the emerging codes. We kept comparing the codes from the data generated. Similar codes were put into subcategories and these subcategories were later put into the main themes. In cases of disagreements, the research team had to discuss until an agreement was reached.

**Rigor.**   To ensure rigor in the data collection, we used Guba and Lincoln criteria [37], that included, data credibility, confirmability, transferability and dependability. Triangulation was checked by team 3 that was devoted to continuous reading through of the transcripts to ensure ongoing comparison of the key information generated from one hospital to another during the data collection and analysis processes. Dependability was observed by the stringent coding procedure and inter-coder corroboration. We made sure to document what each code meant in detail as illustrated in Table 3. Data confirmability was observed by ensuring that participants' statements were captured with barely any modifications made. Data transferability was ensured by the research team so that a rich, thick description of the study process was documented to enable replicability in a similar context elsewhere [40]

## Results

### Characteristics of participants

We interviewed 25 healthcare providers of the different cadres; seven obstetricians/gynecologists, ten nurse-midwives, four nurses and four administrators. Of the 25 interviews, six were in private; ten were in public, while nine were in private not-for profit health facilities. The average age of the participants was 40(±8.7) years. Majority of the participants had Bachelors' degrees and up. The great majority of healthcare providers had more than ten years' experience offering maternal and newborn health services. The socio-demographic characteristics are summarized in Table 2.

The following themes and subthemes were generated from the data analysis. Data comparisons during rigor analysis showed a number of similar experiences in maternal and newborn health service delivery irrespective of the health facility. There were however some disparities in the experiences within the different cadres. Nurses tended to use more of the public means when compared to the obstetricians/gynaecologists and administrators. The way the nurses, navigated through the hassle of transport to the workstations were different from the obstetricians/gynaecologists (Table 3).

**1. Impact of the COVID-19 Pandemic and changes in Quality of Maternal and Newborn health service delivery.**   Irrespective of the health facilities, the discussants reported various

**Table 2.  Socio-demographic characteristics of 25 healthcare providers involved in maternal and newborn healthcare service delivery in public and private health facilities in Kampala, Uganda.**

| Nature of Health Facility | Private(N) | Public(N) | Private Not-for Profit(N) |
|---|---|---|---|
| **Sex** | | | |
| Male | 2 | 2 | 2 |
| Female | 4 | 8 | 7 |
| **Age (years)** | | | |
| 20–29 | 0 | 1 | 1 |
| 30–39 | 4 | 5 | 4 |
| $\geq$ 40 | 2 | 4 | 4 |
| **Cadre** | | | |
| Nurse | 3 | 7 | 6 |
| Obstetrician/Gynaecologist | 2 | 2 | 2 |
| Administrator | 1 | 1 | 1 |

**Table 3. Summary of the themes and subthemes that emerged among healthcare providers on the impact of the COVID-19 pandemic on maternal and newborn health service delivery in public and private health facilities in Kampala, Uganda.**

| Theme | Subtheme | Summary from the quotes |
|---|---|---|
| 1. Quality of service delivery | 1. No change in service delivery.<br>2. Deterioration in service delivery. | • Clinical evaluation didn't change based on COVID-19.<br>• Late presentation by patients, travel restrictions and harassment from security personnel, lack of PPE led to deterioration of service delivery. |
| 2. Lived Experiences during the pandemic | 1. Salary cuts and loss of jobs.<br>2. High patient turn over for the available healthcare teams and burn outs.<br>3. Confrontation from the security personnel during the pandemic. | • Healthcare providers especially in private facilities lost jobs. Some had to take unpaid leaves and pay cuts during the pandemic.<br>• Patient numbers were overwhelming with fewer staffs on ground.<br>• Healthcare providers burnt out and could leave some of the mothers unattended to even in labour.<br>• Healthcare providers were harassed by security personnel as they tried to access their work stations.<br>• Transport to and from work was a hassle for healthcare providers especially the lower cadres.<br>• Limited access to PPE exposing healthcare providers to undue risks of contracting COVID-19 |
| 3. Experiences of dealing with COVID-19 positive patients. | 1. Fear of stigma associated with COVID-19.<br>2. Fear of spreading COVID-19 to family members. | • Stigma was associated with contracting COVID-19. Infected healthcare providers were side-lined for two or more weeks. This meant no earning during the period of self-isolation or quarantine.<br>• The risk of spreading COVID-19 to family members was one of their greatest fears. |
| 4. Motivating factors for the healthcare providers to serve during the pandemic. | 1. Passion to serve, incentives, fear of litigation and need to fend for their families were the commonest motivating factors for the healthcare providers. | • The inherent desire to serve their clients, recognition from the administrators, and fear of litigation kept the healthcare providers going during the pandemic. |
| 5. Healthcare providers' recommendations in service delivery during the current and future pandemics. | 1. Provision of accommodation and transport for healthcare providers during the pandemic.<br>2. Measures to enable healthcare providers reach out to their patients through facilitations like telemedicine, airtime to maintain contact would be so handy<br>3. Assurance of job security during the current and future pandemics.<br>4. Recognition from the government and their employers would keep their spirits high as they serve. | • Healthcare providers suggested that provision of onsite accommodation and transport would enable them serve better.<br>• Databases and telemedicine need to be embraced during the current and future pandemics to ensure continuous service delivery.<br>• Appreciations of their efforts and job security drive their passion to serve. |

opinions of the impact of COVID-19 on the quality of service delivery offered at their different facilities. Some of the healthcare providers reported that there was no change in the quality of services they offered while most of the other discussants reported a tremendous drop in their service delivery following the lockdown, travel restrictions and financial constraints caused by the pandemic. The practice demanded frequent sanitizing, wearing masks and having to undertake examinations like taking a blood pressure measurement, obstetric and general exam of patients in gloves. To some, COVID-19 demanded minimizing physical contact with patients taking out the humane part of the practice. There was a general notion that the measures to contain the pandemic were rushed and that the health system wasn't prepared enough to stand to the COVID-19 pandemic. With the travel bans, both patients and the healthcare providers could not access the health facilities easily. Patients could barely keep their appointments and their late presentations led at times to adverse pregnancy outcomes.

*"The quality of the service went down because of COVID-19. I feel am not giving the mothers the best because many drugs that are out of stock. Then also the number of staff was reduced and those still working are overworked and cannot produce the best, so you find some things not yet done, some treatment not yet given, not because the people forgot but because they are too busy."*

**Obstetrician, Private not-for-profit hospital**

*Failure to keep appointments and its impact on pregnancy outcomes*. Due to financial constraints with closure of businesses and the transport bans, many patients were not able to keep their appointments. This led to many having late presentations at times with dire emergencies. Patients had to collect letters from the Resident District Commissioners (RDC) so as to travel to health facilities otherwise; they would be harassed by security personnel. Even when patients did their best to present to hospitals, the healthcare providers could come in very late and at times totally failing to report to the health facility. There was also closure of some services in many health facilities like antenatal, immunization, postnatal and family planning clinics to minimize on congestion. All these actions had a toll on maternal and newborn health as expressed in the following excerpt,

*"Some patients cancelled appointments because of lockdown and transport. When the President allowed people who are pregnant to move, one of my patients struggled to find ways of getting to the health facility but failed. She lost the baby and we also nearly lost her as well but she pulled through."*

**Obstetrician, Private not-for-profit hospital**

*Increment in the cost of maternal and newborn health services during the pandemic*. Despite the fact that many businesses had to close during the pandemic, the overall cost of care went up especially in private and Private-Not-for Profit health facilities. The patients had to buy masks, sanitizers and also had to undertake COVID-19 screening tests before accessing care at the different health facilities. A COVID-19 screening test was costing between 50–100 USD in Uganda. This led to low patient turn up at the different health facilities and this could have pushed many into the hands of the less skilled traditional birth attendants or home deliveries as expressed in the following excerpt;

*"One of the precautions is every mother coming for an elective or emergency procedure like caesarean section needs to have an unplanned COVID-19 test no later than 72 hours which is very expensive in addition to the hospital costs. Mothers would tell you that we can no longer afford."*

**Obstetrician, Private Hospital**

**2. Lived experiences during the COVID-19 pandemic.**    Salary cuts and job loss. COVID-19 was a nightmare to healthcare providers in Uganda especially those in Private and Private-Not-For Profit health facilities. With reduced patient turn up, the hospital revenues dwindled. This meant that some of the staffs were forced into unpaid leave with others being laid off to survive through the pandemic. Some of the healthcare providers had to take salary cuts with fears of being laid off if they refused to accept the revised contracts. Some of the healthcare providers were the sole breadwinners as their spouses had lost their jobs or businesses during the pandemic. The healthcare providers had to work very long and extra shifts if they were to be paid their fully salaries as expressed in the following excerpt;

*"Yes we worked temporarily without pay then you could be asked to take annual leave then unpaid leave for three months. We didn't know what the future held then; you couldn't afford to take an unpaid leave when you have families to feed at home".*

**Obstetrician, Private not-for-profit Hospital**

*High patient turnover per the available workforce and burnout*. With travel bans and curfew during the lockdown, most of the healthcare providers couldn't access their work stations in time, with most of the healthcare providers living away from their work stations. At some facilities, the healthcare providers were provided transport or accommodated in hostels, however these efforts were unsustainable.

*"COVID-19 indeed affected us for example professionally you are supposed to handover the ward when retiring from the day's work but we cannot sometimes put together a report and handover to the next person because there are many patients and you are very busy. You would leave the patients on the ward without a medical personnel because everyone is rushing to beat the curfew times and the person you handing over to is delayed because they have no transport or they have to walk up to the hospital. One day a mother delivered on the ward in the absence of a healthcare provider you can imagine! Professionally you are not supposed to leave patients unattended to but because of COVID-19 we had no one to blame".*

**Midwife, Public Hospital**

Duty allocations changed from three to two shifts a day and healthcare providers had to start working 12-hour shifts to keep the service delivery running. Some of the clinics had to be changed from offering their services full week to two days a week. This led to burnouts among the healthcare providers with the heaviness of the subsequent clinics leading to patient abandonment that could have led to adverse pregnancy outcomes during the pandemic.

*"Ok, before we used to have two shifts that is day and night and the night person would come at 8:00 pm and leave at 8:00 am because the curfew was before 6:00am and after 7:00pm so by 3:00pm people were leaving and going back. Now we can't handover at 8:00 pm meaning that the night shift will start earlier or the day staffs have to sleepover. Those who used to sleepover overworked because they had to cover up for us who are out. You have few left and the work overload shoots up because the staffs are few and the patients are very many. For example you are 3 staffs on the unit and you have like 8–10 deliveries, so there is a delayed service".*

**Midwife, Private not-for-profit Hospital**

*Confrontation with the security personnel*. Participants irrespective of where they worked reported ugly encounters with the security personnel during the lockdown especially during the curfew hours. They mentioned that they were mistreated and embarrassed yet they considered their services essential to the public in or out of curfew hours as labour is a naturally occurring phenomenon.

*"During curfew time; as they were taking us back home, police officers would stop us and on each roadblock, you would have to show your ID. And of course, the worry that you would have; will I get home safely? There was actually a time they stopped us especially the time when the curfew was at 7 pm and it was coming to 8 pm. They stopped us, we were ordered to come out of the car, and they told us to kneel down. So we all moved out and knelt down".*

**Midwife, Private Hospital**

Some participants reported that if one lacked any identification like work ID, sticker or practicing license, they would be beaten, made to kneel, pay bribes or at worst sleep in police cells. Even with car stickers, the security forces could pull some of the stickers off healthcare providers' cars calling them fake. This left many healthcare providers stranded on how to serve during the COVID-19 pandemic. These encounters inhibited many healthcare providers from turning up for duty especially in the night. The discussants expressed frustrations because they were called for dire emergencies yet they were stopped several times on the way as expressed in the excerpt below;

"*Actually during the lockdown the security operatives would disturb us at the road blocks. They would make you park aside and ask for your identification card or you would sit there and wait for someone to come to your rescue meanwhile there is an emergency. . .even when we got the car passes and stickers, the police men would say they were fake and you would be arrested. We would explain that we were nurses, coming from duty, but the police didn't want to listen*".

**Midwife, Public Hospital**

*Hassle of getting transport to and after work*. With the travel bans, curfew restrictions, the discussants reported that accessing their work stations was a hassle. Irrespective of the health facilities, transit of healthcare providers was complex during the pandemic as expressed above. There were few facilities that had travel plans for their employees. Some of the participants reported walking long distances before reaching the pickup points. The hassle was more among the lower cadres who tended to use public means to access their work stations. Where all means were impossible, healthcare providers were to stay at the health facility as per the President's directive for 14 days to three months.

"*Yes, I struggled to come to work and go back. When I fail to make it to hospital by public means, I used the hospital van which was challenging as it never kept time. You would come from home, stand on the road at 7am, they pick you at 10am and of course you would be late and by the time you reach here, patients have gone away, then again after work you sit and wait until they take you back. It was costly in one way or another. . . sometimes the vehicle would stop in "Kyengera", yet I stay very far but they would call you and threaten you that if you do not come we shall stop you from getting your salary. So I ended up walking*".

**Midwife, Public Hospital**

*Limited supply of personal protective equipment and supply*. It was worth noting that some facilities did not get enough supplies especially in form of personal protective gear. This complicated the work of the healthcare providers hard and thus a barrier to providing maternal and newborn health care services in the different facilities as expressed in the excerpts below.

"*One key aspect is the stock out of the major things we need to use like alcohol [alcohol based sanitizer], gloves actually when COVID-19 set in, we lacked all the essentials masks, sanitizers, gloves so it was a big challenge yet very scary. People saying if you have no mask you are not using PPE and yet you are attending to these mothers you are at risk. So it was a big challenge handling the mothers with all the scares*".

**Obstetrician, Public Hospital**

**3. First encounters with the COVID-19 positive mothers.**   Some of the participants kept fresh memories of their first encounters with COVID-19 patients. The panics of contracting the virus left them with sleepless nights. The fear of having to stay in isolation for 2–3 weeks with no earning haunted many in the private health facilities. This was mainly because some facilities resorted to remunerating staff for days worked as opposed to monthly salaries. COVID-19 was also reported to generate stigma from fellow colleagues. Ambulances with COVID-19 teams donned in scary uniforms could come for those who had encounters with positive patients and would whisk them away. With memories of 15 healthcare providers in Uganda having died from the pandemic, it was inevitable among health workers to fear for their lives. This was compounded by the fact that women in labour were never tested for COVID-19. If they were suspected to have COVID-19, results could only be returned after 72 hours. This was only after the patients paid for the test.

*"The hardest situation I faced was my contact with a COVID-19 patient. Incidentally, my first COVID-19 patient was a healthcare provider who happens to be working from a different hospital within Kampala and she had worked on a COVID-19 patient a few days earlier. She had been tested but when she came she didn't inform us, we worked with her and the following day after working, her results came out positive. So I had to be quarantined, the nurse who received the baby and the patient that we worked on. All those we had worked with had to be isolated and I didn't like the way my patient was handled by my colleagues that stayed behind because I realised a lot of stigma had been left behind. People feared to touch that lady, people feared to give treatment, nurses were so scared and this was during the lockdown. The patient wasn't treated well at all mainly because of the fear and the stigma and it's still on up to now."*

**Obstetrician, Private Hospital**

*Facing the greatest fear of contracting COVID-19.* With barely any life insurance policy for private and public health facilities in place, the greatest fears that almost all the participants acknowledged and reported was the fear of contracting COVID-19. This is because most of the participants did not know much about the disease and were fearful of contracting it and taking it to their families. This was the greatest nightmare for all health workers. The inability to provide for their families, the psychological torture and the loneliness that came along with the isolation bothered most of the discussants as the following excerpts show,

*"Sometimes we don't have PPE so for me if the blood flashes on me I have a hundred questions to myself. Am I really safe? Yet you are also scared to go for the tests. Actually, am told that COVID-19 is also scaring and the disease leaves the scars in your lungs then you already know that you have a few days to live".*

**Midwife, Private Hospital**

*"On my side, contracting COVID-19 while offering the service is my greatest fear because I go back home—contracted COVID-19 or not I don't know but then I interact with my family and those at home. They can all contract the virus which is a problem and still on my side".*

**Midwife, Private Hospital**

**4. Motivating factors to endure through the COVID-19 pandemic.**   Amidst all the challenges posed by the pandemic, healthcare providers found various motivating factors to serve Ugandans. Some of the discussants highlighted that the following motivating factors kept them going; passion to serve which was beyond their salaries, recognition from their colleagues

and administrators, teamwork, financial and other incentives from the government or their employers, the attachment they have with their clients, the inherent responsibility to provide for their families, serving being viewed as a gesture of patriotism, fear of litigation when the mother or baby's lives were in jeopardy and the joy of having a healthy mother, baby and a happy family as expressed in the following excerpt.

*"First of all, the corporation we have, the gynaecologists are there and they are good people. You work as a family so when you work as a family, I think you are always happy to come in the morning and say I think am going to work in a place. Am happy and I have peace and go back home well, I think that's one of the best things we have".*

**Midwife, Private-Not-for Profit Hospital**

**5. Suggestions to improve maternal and newborn service delivery during pandemics.**
Having encountered the current COVID-19 pandemic, the discussants had a number of suggestions they put across to streamline maternal and newborn service delivery to minimize on the adverse outcomes that were posed by the pandemic. It was vivid that there were hardly any systems in any of the health facilities, to trace and reach out to their clients remotely (telemedicine) to minimize on congestions.

During the pandemic, most of the Continuous Medical Education (CME) meetings were stopped and the healthcare providers were left to battle out COVID-19 on their own as platforms like zoom meetings were not accessible to most of the healthcare providers. Healthcare providers at the frontline of maternal and new born health, felt less appreciated by the government because they were not directly involved in active management of COVID-19 patients. The discussants made the following recommendations, future investments in the digital medical services to facilitate on going medical education to the healthcare providers, incentives and risk benefits to be given to healthcare providers at the frontlines, provision of onsite accommodation, readily available drugs, equipment, supplies and PPE for health workers to improve their service delivery, door-to-door healthcare services to be facilitated to minimize on delays like in immunization, antenatal and postnatal clinics, the need to invest in hospitals having databases and means of communication to reach out to their clients during pandemics. Healthcare providers desired to have job security even during pandemics, availability of transport means other than ambulances to enable easy access to their work stations. The discussants demanded respect from the various security agencies because of the relevancy of services they offered to Ugandans was a gesture of patriotism. Some of the major recommendations included the following;

*Resource facilitation to contact clients.* Some of the participants reported the need for facilitations to do with communication services so that they could get in touch with their patients. This would make it possible for the healthcare providers to know how the patients were and how to organise ambulance services in real time. In so doing, there would be improvement in service delivery and patient management before any complications could worsen.

*"We should be facilitated with airtime to maybe call our clients and encourage them and also contact their peers because they are also there.*

**Midwife, Private Hospital**

*Door-to-door services during pandemics.* Most of the participants also recommended outreach services in future if such a pandemic happens. This was after realizing that most patients missed their visits while others got complications from home. Participants believed that if

services were taken closer to the people, it was going to make people's lives better and also reduce tremendously the complications related to delayed service delivery.

*"One of the key recommendations is to ensure that even in such times, the pregnant mothers can access services and health facilities at any time they can be given those toll-free numbers to call an ambulance to pick them up. If you were keen on the news, many mothers died and or lost their babies during the lockdown period because of transport. If may be the government can extend maternal services to all nearby health facilities whether Health centre II or III, that can help so the mother can access a service easily because during pandemics accessibility is the main issue".*

**Midwife, Private-Not-for Profit Hospital**

*Improving staff access to health facility/ providing transport.* Provision of staff with transport was recommended for all the health facilities. This was because what was provided this time was not comprehensive enough and a more robust mode of transport was recommended. This was also after a realization that not all the healthcare providers can be accommodated at the facilities yet all services have to continue.

*"But I suggest that during pandemic times the hospital should be more prepared to reach its staff and bring them to work especially those who come from far. I know you cannot make all the healthcare providers reside nearby- many have families to take care of. …. Because if you receive a dying mother and you are trying to save her, you need the other colleagues–like consultants to be able to come to your rescue as soon as possible. The doctors and nurses are there and when we fail we refer".*

**Administrator, Private-Not-for Profit Hospital**

*Availability of personal protective equipment and supplies.* Majority of the participants also recommended equipping all the health centres with enough supplies. This was because there were rampant shortages in personal protective equipment in a number of the health facilities which was putting the lives of the healthcare providers and their patients in jeopardy.

*"The health facilities should then be equipped with all the necessary supplies for obstetric emergency cases depending on the level of the health facility. Also we don't normally get such big pandemics but this has taught us to be prepared and the hospitals need to be very prepared for such situations—have all the necessary equipment ready and on standby".*

**Midwife, Private-Not-for Profit Hospital**

*Addition of the COVID-19 testing to routine laboratory workup.* COVID-19 testing was also recommended to be part of the routine at the health centres. This is because most of patients seeking maternal and newborn health services were not screened for COVID-19. The mothers who came in labour were not tested, thereby putting the lives of the healthcare providers and fellow mothers at risk of contracting COVID-19.

*"I think for a mother coming to the labour ward, it should be a must that they are tested for COVID-19 just like we do for HIV and I think it should get in the routine because you never know whom you are dealing with. A mother may come when she is pregnant and you think that she is safe and yet she is not.*

**Midwife, Private Hospital**

*Provision of staff accommodation at the health facility*. Most of the participants also recommended housing facilities.

*"I would recommend hospitals to put up many hostels where healthcare providers can stay during time of pandemic and it should be near the health facility. During the lockdown, we had very few rooms for accommodation but most staff were coming from outside and took a long time to arrive and had to leave earlier".*

**Midwife, Public Hospital**

## Discussion

This study sought to understand the experiences and perceptions of healthcare providers at the frontline as they offer maternal and newborn care services in both public and private health facilities in the first wave of COVID-19 in Uganda with the aim of streamlining patient care in face of current COVID-19 pandemic and in future disasters.

The discussants elaborated that there were transport challenges irrespective of the cadre and access to their work stations was through much hassle. It was vivid that most of the public, Private-Not-for Profit and private health facilities lacked accommodation for their staffs. Work overload and burnout, inadequate supply of personal protective equipment, drugs and supplies were common. Healthcare providers reported being harassed by the security agencies; they felt unappreciated by their different employers. Memories of their first encounters with COVID-19 patients, loss of employment, fears of contracting COVID-19, fears of being in isolation, or quarantine with inability to fend for their families were the worst experiences in their careers. Passion to service, incentives and appreciation from their employers, availability of transport and accommodation were the major motivators among healthcare providers offering maternal and newborn services in private and public health facilities in Kampala, Uganda.

Similar trends of patient overload, long waiting times, inability to keep appointments and cancellation of antenatal, postnatal and immunization clinics have been reported by Hussein et al during the COVID-19 pandemic in other low resource settings [41].

From prior pandemics like the flu pandemic caused by swine influenza (H1N1), over 50% of the healthcare providers contracted the virus according to Stephens [42]. There is a likelihood of a similar trend in Uganda having had already 37 healthcare providers succumbing to the deadly COVID-19 [30]. This could be the reason behind the health providers' greatest fear of contracting the COVID-19. With barely any efforts in place to screen for COVID-19 (real-time quantitative reverse transcription PCR (qRT-PCR) [43], among clients seeking maternal and newborn health services, this nightmare could become a reality in Uganda. When an effort to undertake a COVID-19 test by the patients was made, the results could take between 24–48 hours to be released whether in private or public institutions [44] yet the test currently costs 60 USD [45]. This cost is almost two thirds of the monthly earning of an employed average Ugandan [46, 47]. With the business closures during the lockdown, very few Ugandans could afford the COVID-19 test. This could put many Ugandan healthcare providers at risk of contracting COVID-19 especially with the scarcity of Personal protective equipment in most of the health facilities [6, 48].

Lessons from the Ebola outbreak in West Africa show that there was a reduction of 27.6% in service use and 44.3% decrease in inpatient services in high incident areas [49], while in Taiwan in 2003, during the Severe Acute Respiratory syndrome, there was a 23.9% reduction in ambulatory care and a 35.2% reduction in inpatient care [50]. The Ugandan government

needs to embrace such lessons to attract its invaluable human resource the healthcare providers into the health facilities if Ugandans are to maintain their trust in the health system. Failure to do so might lead to Ugandans seeking care from alternative avenues which could be rather unsafe leading to adverse maternal and newborn outcomes. Evidence shows that when the healthcare providers offered their skilled attendance at birth such adverse outcomes can be averted [51, 52] thereby enabling Uganda achieve the UN Sustainable Development Goal 3 of less than 70 maternal deaths per 100,000 live births [53].

With the travel bans, curfew and lockdown in place, healthcare providers struggled to access their work stations [6, 7] with some of them having been hurt in the process. As highlighted by the discussants, there's a need for the government to mediate a friendly working environment between the security personnel and the medical fraternity to minimize on delays seen during the pandemic that could have grossly affected maternal and newborn service delivery like immunization, health facility deliveries, antenatal and postnatal clinic attendance [6–8, 52]. The World Health Organization recommends that all stakeholders should ensure that despite the physical (the lockdowns and curfews), financial (unemployment, financial losses) and social (fear of contracting COVID-19 from health facilities, social distancing) barriers, measures of transporting healthcare providers to their respective health facilities should be streamlined during the pandemic. The government of Uganda could lobby for funds from the World bank and other funders to ensure continuity of maternal and newborn health care services during the pandemic [16, 20].

Even in resource rich settings like United States, healthcare providers have been motivated with bonuses and incentives in addition to the readily available personal protective equipment at all levels since healthcare providers are invaluable in controlling and managing the COVID-19 pandemic [54]. If at all the cries of the healthcare providers at the frontlines are addressed [7], the level of service delivery could improve thereby improving the quality of care in maternal and newborn health. According to the discussants, availability of transport means and accommodation in most of the Ugandan facilities could equally improve service delivery and optimize patient outcomes during the current pandemic and in future disasters [6, 55].

As mentioned in most of the interviews, there's a need for the government to invest in Health management information systems and digital medical services, with lessons learnt from the current pandemic, to enable healthcare providers to reach out to low risk patients in the community and streamline patient referrals. In so doing, congestion at health facilities could be mitigated [56]. The interaction between healthcare providers and patients could be improved thereby enhancing doctor-patient relationships and optimizing patient outcomes. This would also improve the patients' trust in the health system [57]. The World health Organization as well recommends the use of mobile services and Tele-health mechanisms for service delivery and training during the COVID-19 pandemic to minimize congestion of health facilities. In so doing the community will be provided with information also to demand for health services with different measures in place to maintain the trust of the community in quality health care. This will also ensure timely health seeking behavior thereby reducing on the preventable maternal and newborn deaths that come along indirectly with the COVID-19 pandemic [20].

As noted in other low resource settings, there's need for strategic planning and simplified governance in Uganda so as to mitigate the health system collapse to ongoing maternal and newborn health services while containing the COVID-19 pandemic [16]. As highlighted by the discussants, to minimize on burn outs of the limited human resource, frequent stock outs, measures such as task shifting, deployment and maintenance of the essential drugs, infection control practices need to be streamlined from the outset by the Ministry of Health

and other stakeholders [20]. Every COVID-19 response team ought to have a member from the maternal and newborn service delivery committee to guide on how these services will be maintained during the pandemic. This calls also for measures to prevent diverting funds and the key maternal and newborn health human resource to maintain equitable access to these services.

The strengths of this study is that we were able to conduct 25 in depth interviews among healthcare providers of different cadres in the eight most utilized private, private-not-for profit and public health facilities offering maternal and new born health services in Kampala, Uganda. This enriched the data collected as healthcare providers at different levels of service delivery were interviewed independently. We were able to achieve data saturation by the end of the data collection. The interviews were conducted in quiet and safe rooms observing the COVID-19 guidelines. This environment was ideal for participants to express their views without any fears or intimidation from colleagues. We were also able to use experienced research assistants to collect the data. To ensure trustworthiness, two independent researchers compared field notes with the transcripts. This is one of the first studies locally that has assessed the lived experiences among health care providers offering maternal and newborn health services. Most of the documented experiences have been in local media once in a while. This will therefore give a more comprehensive opinion on the overall lived experiences among healthcare providers nationwide.

From the discussions, there were few healthcare providers that were directly involved in COVID-19 patient management. Involving more of these healthcare providers in the interviews would have enriched the study findings. This study could have been more interesting if data collection from the healthcare providers was conducted concurrently with patient interviews. Some of the themes generated in the patient interviews could have enriched the healthcare providers' interview guide to assess the perspective of the patients in regards to quality of care they were receiving during the COVID-19 pandemic with what the healthcare providers regarded as quality care. Review of the health facility data records could have made our study more informative of the actual trends in service delivery during the COVID-19 pandemic as discussed in the interviews. We also acknowledge that involvement of healthcare providers in far to reach areas would have made our study findings more informative of the overall experiences of the medical fraternity in the current pandemic.

## Conclusion

The COVID-19 Pandemic has led to a decline in quality of maternal and newborn service delivery by the healthcare providers as evidenced by the shorter consultation time, closure of vital clinics and elective surgeries, longer waiting times, failure to keep appointments and shortages in medical supplies and PPEs. The major hurdles to service delivery in the first phase of the lockdown included harassment from security agencies, fear of contracting COVID-19, loss of employment, failure to access transport or accommodation at their work stations. The healthcare providers despite the limitations imposed by the pandemic were driven by the inherent passion to serve their patients, availability of transport means, and appreciation from their employers. The African governments need to address the raised concerns of healthcare providers to improve service delivery and prevent burnouts during the current and future pandemics.

Availability of accommodation and transport means at the facilities, provision of PPE, bonuses and inter professional teamwork are low lying fruits that needed to be tapped to drive teams during the current and future pandemics.

## Supporting information

**S1 Dataset.**
(DOCX)

## Acknowledgments

We indebted to the research team and study participants for making this research project a reality. With great pleasure we appreciated the awesome input of Richard Muhumuza and Andrew S. Ssemata in critiquing the interview guide, data collection and analysis. A special vote of thanks goes to the Administrators of the health facilities that participated in the study. We are grateful to the administrators of the Mak RIF project at Makerere University for the all the help in securing the funds for this project.

## Author Contributions

**Conceptualization:** Herbert Kayiga, Pauline Mary Amuge, Racheal Samantha Nanzira.

**Data curation:** Andrew Sentoogo Ssemata.

**Formal analysis:** Herbert Kayiga, Andrew Sentoogo Ssemata, Annettee Nakimuli.

**Methodology:** Herbert Kayiga, Racheal Samantha Nanzira, Annettee Nakimuli.

**Project administration:** Diane Achanda Genevive.

**Resources:** Racheal Samantha Nanzira.

**Supervision:** Herbert Kayiga, Diane Achanda Genevive, Pauline Mary Amuge, Annettee Nakimuli.

**Validation:** Andrew Sentoogo Ssemata.

**Writing – original draft:** Herbert Kayiga, Pauline Mary Amuge.

**Writing – review & editing:** Herbert Kayiga, Diane Achanda Genevive, Pauline Mary Amuge, Andrew Sentoogo Ssemata, Racheal Samantha Nanzira, Annettee Nakimuli.

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
