## [Decision Letter · Decision Letter 0]

30 Apr 2021

PONE-D-21-03659

Lived Experiences of Frontline Healthcare Providers offering Maternal and Newborn Services amidst the Corona virus Disease 19 Pandemic in Uganda: A Qualitative study

PLOS ONE

Dear Dr. Kayiga,

Thank you for submitting your manuscript to PLOS ONE. After careful consideration, we feel that it has merit but does not fully meet PLOS ONE’s publication criteria as it currently stands. Therefore, we invite you to submit a revised version of the manuscript that addresses the points raised during the review process.

Thank you for this submission that includes important time-sensitive data. Both reviewers have provided detailed comments focusing on the introduction, methods, and discussion that give practical recommendations to improve the readability and length of the manuscript. Please address each reviewer's comment carefully.

We look forward to receiving your revised manuscript.

Kind regards,

Michelle L. Munro-Kramer, PhD, CNM, FNP-BC

Academic Editor

PLOS ONE

Journal Requirements:

Please include a copy of the interview guide used in the study, in both the original language and English, as Supporting Information, or include a citation if it has been published previously

The authors have declared that no competing interests exist.

We note that one or more of the authors are employed by a commercial company: UbunifuAfrika Limited

Reviewers' comments:

Reviewer's Responses to Questions

**Comments to the Author**

1. Is the manuscript technically sound, and do the data support the conclusions?

Reviewer #1: Yes

Reviewer #2: Partly

2. Has the statistical analysis been performed appropriately and rigorously? 

Reviewer #1: N/A

Reviewer #2: N/A

3. Have the authors made all data underlying the findings in their manuscript fully available?

Reviewer #1: Yes

Reviewer #2: No

4. Is the manuscript presented in an intelligible fashion and written in standard English?

Reviewer #1: Yes

Reviewer #2: Yes

5. Review Comments to the Author

Reviewer #1: Thank you for the opportunity to review, “Lived Experiences of Frontline Healthcare Providers offering Maternal and Newborn Services amidst the Corona virus Disease 19 Pandemic in Uganda: A Qualitative study.” The paper is very interesting and highlights the complex environment in which MNCH providers have been operating with the COVID-19 pandemic. I believe this article will be very helpful to the readership of PLoS One and illuminate the challenged MNCH providers face. I only have two major comments for consideration and a few minor points.

1. While very interesting, the paper needs to be overall streamlined and synthesized. The quotes are helpful to understand the nuance, but each theme does not require 3-5 quotes. The authors should revise the paper and messaging to ensure readability, appropriate length, and synthesis. One potential approach to do this in the Methods is put the text description of the 8 hospitals in a table to streamline the text (i.e. remove lines 149-213). Similarly consider the quotes for each theme.

2. Depending on what was agreed to as part of the informed consent process and confidentiality agreement with the IRB and participants, the authors need to be very careful about identifiability of participants. Based on Table 1 alone, it is not unreasonable to expect that an individual clinician could be identified and the quote attribution. Given that the hospital is named and the cadre identified, it would be relatively easy to identify the interviewed person and potential put them at risk and violation of privacy/confidentiality. Further, since the authors described harassment of the providers by security forces, it would be even more important to protect participants from any retribution for speaking up. A few suggestions to remedy this: (a) Summarize Table 1 into descriptive categories of the 25 individuals (i.e. X were obstetricians, Y had Diplomas, Z were married); (b) Remove specific facility names and any names of persons (or identifiable traits) from the quotes, for example, “The Bishop of Namirembe Diocese managed to 659 get us some stickers. Maybe for your information, the Bishop is the patron for the 660 hospital. This is an Anglican founded hospital so that’s why the bishop had to come in.” (Line 658-660).

MINOR

1. Can the authors describe the duration of lockdowns in Lines 90-95?

2. Please clarify in Line 88 is these are only COVID-19 tested and confirmed cases? Are there estimates of suspected case load that could be described?

3. Is there documentation of impact on MNCH attendance?

4. In the methods, then authors note this was part of a “bigger” study? Please clarify in the methods that this was an embedded qualitative study. Also, can the authors describe how they selected the 25 participants?

5. Was any software used for data analysis?

Reviewer #2: 1. The manuscript is probably technically sound but clarifications are needed in the methods to confirm this statement.

3. The authors state that they have made all the data available but it is obvious that they have not nor could they do this. This is a textual data set consisting of direct transcripts of many hours of interviewing. Making it available would violate the subjects' confidentiality and privacy. Best to state this in the manuscript that saying that all the data have become available.

4. The results and discussion sections are presented in an intelligible fashion but the intro and methods are not. The reviewer suggests how to improve in the comments that follow.

Additional comments by the reviewer are provided below:

1. The strength of the manuscript lies it its reported data.

2. Please, increase the readability of the data in two ways:

a. Improve the quality of writing in the intro and methods so that the reader has stamina left to make it to the results. This reviewer suggests ways to do that in more detail further down in their review.

b. Improve the readability of the results section by:

i. Creating a table of contents of the themes and subthemes that emerged from the data; state in one sentence a summary of what was found in each theme or subtheme; and the page number where the theme or subtheme is described. You currently have over 13 different themes, with subthemes under some of the themes – it is impossible for the reader to follow. Please number all themes on this table and in the narrative.

ii. Going over each theme and thinking of ways to shorten either the text or the length of the direct quote to convey only what is said in a unique way that the text has not said. Current length is over 23 pages – bring it down to 12 pages.

3. Safeguard the anonymity and confidentiality of the data by removing the names of Hospitals and the age of each discussant from Table 1. For the purposes of the publication, the categories that you mention of “two Private hospitals, three Private-Not-for Profit hospitals and three Public health facilities” are sufficient for the reader to understand the study context. Replace hospital name by Private hospital 1, Private hospital 2, Private not for Profit hospital 1, Private not for Profit hospital 2, and so on. You report the mean age and range in the text so no need to specify on the Table. Instead of nurse 1, nurse 2, report the number of nurses, admins, and other providers interviewed in each hospital on the Table. By doing these changes you also increase the readability of current Table 1.

4. Methods

a. Describe all the teams that were involved in the study in one place. It looks like you had a data collection team, a data analysis team, and a team of external or independent (unclear) investigators that was checking the rigor. Like this: Team 1 had this and that investigator and did this; Team 2.. and so on.

b. For rigor, you examined the credibility, confirmability, transferability and dependability of the data; also their congruency, triangulation and trustworthiness. These are all noble goals but your description of each is so confusing to any reader who might like to replicate your work at their hospitals, to the point of giving up reading the manuscript right there. You may clarify that these are related but slightly different aspects of data validity and you made sure to check for all of them by having a separate team constantly comparing the field notes, the transcripts, the codes, and their interpretation. Your manuscript does not describe any other evidence to compare your data with, neither do you ever mention that you took videos which brings on another layer of ethics and IRB approval issues. Please clean up and shorten that section.

5. I will now further elaborate the changes needed in the intro that I mention under comment 2.a.

1. Ref 1 – you need a HWO reference for this

2. Ref 2 – need full citation (page numbers, issue)

3. Ref 3 need the complete citation, weblink, and when it was accessed

4. Ref 4 , 5 – same

5. “with many vulnerable 90 populations, 1.4 million HIV positive patients, 800,000 diabetic patients and 100,000 TB sputum 91 patients, the President of Uganda, Yoweri Kaguta with his cabinet initially closed public 92 gatherings, shopping malls, and public domestic and international travels from the 18th March 93 2020.”

a. The first part of this sentence belongs in the next paragraph on how the health care system in Uganda was already overwhelmed, and needs the citation of ref 9. The second part of this sentence doesn’t need the name of the President for a scientific publication but rather a reference from a government website.

b. Use a better term for TB sputum patients

6. Ref 6 and 7. The reader needs to know the full citation. Is this a scientific journal? A newspaper? Provide the weblink and when it was accessed.

7. Ref 8 should be a ref on Uganda prior to the pandemic – this a WHO reference on the pandemic. Best to only refer to 9 there (line 100). Ref 8 is appropriate in line 107.

8. “The Ministry of Health in Uganda has already reported a decline in the current immunization coverage 104 though the overall impact of 105 COVID-19 on the immunization coverage is yet to be determined.” Need a citation or rephrase.

9. Ref 6 and 12 are the same. Streamline.

10. Ref 11. Need to know the type of document, weblink and date accessed.

11. Ref 13. The reader needs to know if this is a book, a journal, a report, the weblink and when it was accessed.

12. “In Uganda, 4,600 women deliver everyday” rephrase – perhaps you mean to say that there are xx deliveries per day.

13. “Interruption in access to quality maternal 108 and newborn health services could put over 10,000 lives of these women and their babies in 109 danger.” Show your calculations and define the time period.

14. “Due to shortages in the personal protective equipment in a number of health facilities, many 112 health workers offering maternal and newborn services fear for their lives.” This sentence already gives your study results away. It belongs to the results section. Delete from intro. Write it in the past tense in the results.

15. “some health workers can’t access their work stations easily 118 with the COVID-19 travel restrictions” This gives away the data – put in the results section. Use past tense in your writing throughout to avoid being outdated and to maintain tense sequence throughout the manuscript.

16. Format of ref 15 and 16 is good. Make all refs read like this.

17. “Some reports reveal health workers having been assaulted 119 by security personnel as they try to access or leave their work stations especially during curfew 120 hours [12, 15, 16]. “ Define security personnel. Do you mean police, hospital security, private firms? But this sentence also gives away your results – best to delete and let your data speak. Again, use past tense.

18. Ref 15 is about women in labour not health workers. Use an appropriate reference.

19. “some health workers have further have been deployed to manage the COVID-19 patients [9].” You cannot use a 2016 reference to substantiate a statement about 2020 or 2021. Also fix the grammar.

20. “with the aim of streamlining patient care in similar future disasters.” This is a good aim, but you also need an aim on streamlining patient care during the current disaster which still lingers in a big way. Edit abstract, intro and discussion on this.

6. Here I elaborate the changes needed in the methods under comment 2.a.

1. Reference the larger study, even if only manuscript in preparation

2. “phenomenological and inductive thematic approaches” Need a ref for phenomenological and for inductive.

3. ““Quality was determined through retrospective 136 review of hospital records on maternal and newborn services that included; hospital deliveries, 137 antenatal attendances, immunization coverage, family planning services offered, postnatal clinic 138 attendance, HIV care services six months before and during the COVID-19 pandemic. Using 139 interviewer-administered questionnaires; we also collected patient quantitative data to assess any 140 trends in the care offered during the COVID-19 pandemic.” You do not need this statement. It refers to a different study. Delete.

4. “We opted for in depth interviews 141 instead of focus group discussions to ensure social distancing and minimize spread of the 142 COVID-19.” There are certain strengths that focus groups have as a data collection method compared to in depth interviews, and vice versa. Say what you missed by not conducting focus groups due to the reasons that you cite and what you gained by conducting IDIs. The statement as is shows lack of understanding of focus groups as a qualitative research method.

5. “We conducted 50 in depth interviews. Twenty five among pregnant and 143 breastfeeding women to assess their lived experiences, perceptions about the quality of services 144 offered to them during the pandemic with the ultimate goal of identifying gaps and what their 145 level of satisfaction was as they sought care during the COVID-19 pandemic.“ This sentence does not belong to this manuscript. Here describe the methods you used only for this manuscript, i.e. the 25 interviews with health providers.

6. Ref 17 needs completion. I will now stop repeating this comment. Please check all refs and provide the weblink and when it was accessed.

7. Ref 18, 19, 22 and 24 are not needed.

8. “merging” you mean “emerging”

9. “The 261 interviews were tape recorded and transcribed verbatim” check and eliminate repetitions. This phrase appears under data collection, under quality control, and under data analysis. The recording should be mentioned under data collection only. The transcription under data analysis only.

10. “To ensure trustworthiness and credibility, two independent researchers read and reviewed the content (interview transcripts, 290 and field notes word-for-word, line-for291 line) several times. Field notes and interview transcripts from each of the interviews were 292 assessed individually and later integrated to strengthen the data analysis and dependability of the 293 study findings. “This is stated under data analysis but under Data collection it is also stated that “All of the interviews were tape recorded and 255 transcribed verbatim immediately thereafter. Transcription accuracy was ensured at the end of 256 the interviews by the Principal investigator and one Administrator. Field notes and the 257 transcription were compared for congruency.” Best to eliminate from both data analysis and data collection sections and describe in simple words as I suggest in 4.b under Rigor only.

11. “We 307 thereafter identified themes reflecting on the depth. We later compared with other classes so as to 308 delimit the theories and achieve conceptual congruency[31].” Explain to the reader what type of depth you reflected on, and what were these other classes and theories that you mention. Better yet, delete.

12. “We observed data credibility by 312 ensuring checks by two members of the research team to accept codes from the transcription” This you did for confirmability. The credibility was already established because you had the recordings to prove it. Delete and summarize as I suggest in 4.b.

13. “peer debriefing from the senior researchers” unclear what this did – probably transferability on whether the data from one hospital was similar for another hospital? Clarify or delete and summarize as I suggest in 4.b.

14. You mention an “external investigator” in line 324 and independent investigators” in line 327. Were these investigators different from the PI, and admin? Best to delete whole section and follow my suggestion under 4.a.

15. Results: Lines 330 to 333 are repetitive - delete. Start section by saying that Table 1 describes the types of cadres interviewed… and edit Table 1 according to my suggestion 3.

16. Results: “The participants 342 generally had a number of similar experiences in regards to maternal and newborn health service343 delivery irrespective of the nature of health facility they worked in.” Since you took such great efforts on rigor, it should have been those multiple checkers who said that the data were comparable among all hospitals in the study and that the participants had similar experiences which allowed for their collective reporting. State that in the results, with the exception of your first theme on preference for working in Private, not for Profit or public facility, where the lived experiences differed.

END OF COMMENTS

6. PLOS authors have the option to publish the peer review history of their article (what does this mean?). If published, this will include your full peer review and any attached files.

Reviewer #1: No

Reviewer #2: No

---

## [Author Response · Author response to Decision Letter 0]

28 Jun 2021

AUTHORS’ RESPONSE TO REVIEWS

TITLE: Lived Experiences of Front line Healthcare Providers offering Maternal and Newborn Services amidst the Novel Corona virus Disease 19 Pandemic in Uganda: A Qualitative study

Authors

Herbert Kayiga (hkayiga@gmail.com)

Diane Achanda Genevive (achandadiane@gmail.com)

Pauline Mary Amuge (paulacallista@gmail.com)

Andrew Sentoogo Ssemata (andrewssemata@yahoo.co.uk)

Racheal Samantha Nanzira (nanzirasamanthar@gmail.com)

Annettee Nakimuli (Annettee.nakimuli@gmail.com)

Version: 2

Date: 7th June 2021

Authors’ response to reviews: See over

 7th June 2021

TO: THE PLOS ONE EDITORIAL TEAM

Object: PONE-D-21-03659 Lived Experiences of Frontline Healthcare Providers offering Maternal and Newborn Services amidst the Corona virus Disease 19 Pandemic in Uganda: A Qualitative study

With great pleasure we are thankful for your consideration of our manuscript for publication in your reputable journal. In response to the editors’ comments sent to us on 30th April 2021, we have revised the above manuscript accordingly. 

Journal Requirements:

 Response: The manuscript has been edited as advised according to the journal’s recommendations.

2. Please include a copy of the interview guide used in the study, in both the original language and English, as Supporting Information, or include a citation if it has been published previously

 Response: The interview guide was in English. It was as follows;

HEALTH WORKERS’ QUALITATIVE INTERVIEW GUIDE:

Qualitative study among health care providers (HCPs) on the Impact of COVID-19 on Maternal and Newborn Healthcare Service delivery

Aim: To understand the lived experience of Healthcare providers as they offer maternal and newborn service during the COVID-19 pandemic, including perceptions on provision of their service in resource constrained settings. 

Research questions:

1. How do HCPs perceive maternal and newborn health service provision in crisis like the ongoing COVID-19 pandemic?

2. What are the most important barriers they face while providing maternal and newborn health care with the ongoing COVID-19 pandemic? 

3. What are the most important facilitators/motivators that get them going as they offer maternal and newborn healthcare service during the COVID-19 pandemic?

Interview Guide questions:

1. Please tell me a little about yourself (Date of birth, education background, marital status, religious affiliation, occupation, number of children, health facility you attended, number of years at the facility etc.)

2. What has been your experience of work before, during the lockdown and after the lockdown and COVID-19 pandemic? Was there anything different i.e. transport, other forms of facilitation. Did anything change in the way they used to do their work? What changes occurred. Also Ask how has your practice changed or been affected by the lockdown and COVID, What are you doing differently. 

3. What precautions are you taking and giving to the mothers, How has this affected your work, How did COVID-19, lockdown affect the quality of service

4. What are the most important facilitators/motivators as you offer maternal and newborn healthcare service during the COVID-19 pandemic?

5. What have been the most challenges that you have faced as you care for mothers and their newborns during this period? Probe for staffing, drugs PPEs and other necessary supplies, access to facilities etc.

6. What are or were your greatest fears as you offer serve at the frontline as a maternal and newborn healthcare provider?

7. What would you recommend to be improved in maternal and newborn health as health workers to enable you serve your clients better in times of pandemics like COVID-19?

The authors have declared that no competing interests exist.

We note that one or more of the authors are employed by a commercial company: UbunifuAfrika Limited

 Response: Thanks for this concern. Diane Achanda Genevieve had used the address of her husband as she was changing employment during the study period. She has never worked otherwise under Ubunifu Afrika Limited. She’s currently a Nutritionist at Kawempe National Referral Hospital. Her address has been changed accordingly. 

 Response: Thanks for the advice. The funding organization had no role in the study design, data collection and analysis, decision to publish, or preparation of the manuscript and only provided financial support in the form of research materials. This has been updated in the Author contribution section. 

 Response: The address of Ubunifu Afrika Limited was just used by one of the authors as a contact address but we have no dealings of any regard with this company in our study.

 Response: The address of Ubunifu Afrika Limited was just used by one of the authors as a contact address but we have no dealings in any regard with this company in our study. We therefore declare that Ubunifu Afrika Limited does not in any way alter our adherence to any of the PLOS ONE policies on sharing data and materials.

Response: We declare that we have no competing interests of any sort whether financial or non-financial.

Response: The address of Diane Achanda Genevieve has been changed as shown in line 17. The other authors and their affiliations are unaltered. 

Reviewers' comments:

Reviewer's Responses to Questions

Comments to the Author

1. Is the manuscript technically sound, and do the data support the conclusions?

Reviewer #1: Yes

Reviewer #2: Partly

2. Has the statistical analysis been performed appropriately and rigorously?

Reviewer #1: N/A

Reviewer #2: N/A

3. Have the authors made all data underlying the findings in their manuscript fully available?

Reviewer #1: Yes

Reviewer #2: No

4. Is the manuscript presented in an intelligible fashion and written in standard English?

Reviewer #1: Yes

Reviewer #2: Yes

5. Review Comments to the Author

Reviewer #1: Thank you for the opportunity to review, “Lived Experiences of Frontline Healthcare Providers offering Maternal and Newborn Services amidst the Corona virus Disease 19 Pandemic in Uganda: A Qualitative study.” The paper is very interesting and highlights the complex environment in which MNCH providers have been operating with the COVID-19 pandemic. I believe this article will be very helpful to the readership of PLoS One and illuminate the challenged MNCH providers face. I only have two major comments for consideration and a few minor points.

1. While very interesting, the paper needs to be overall streamlined and synthesized. The quotes are helpful to understand the nuance, but each theme does not require 3-5 quotes. The authors should revise the paper and messaging to ensure readability, appropriate length, and synthesis. One potential approach to do this in the Methods is put the text description of the 8 hospitals in a table to streamline the text (i.e. remove lines 149-213). Similarly consider the quotes for each theme.

Response: Thanks for the advice. A table has been inserted and has the descriptions of the eight facilities. We have deleted lines 149-213 as recommended by the reviewer.

2. Depending on what was agreed to as part of the informed consent process and confidentiality agreement with the IRB and participants, the authors need to be very careful about identifiability of participants. Based on Table 1 alone, it is not unreasonable to expect that an individual clinician could be identified and the quote attribution. Given that the hospital is named and the cadre identified, it would be relatively easy to identify the interviewed person and potential put them at risk and violation of privacy/confidentiality. Further, since the authors described harassment of the providers by security forces, it would be even more important to protect participants from any retribution for speaking up. A few suggestions to remedy this: (a) Summarize Table 1 into descriptive categories of the 25 individuals (i.e. X were obstetricians, Y had Diplomas, Z were married); (b) Remove specific facility names and any names of persons (or identifiable traits) from the quotes, for example, “The Bishop of Namirembe Diocese managed to 659 get us some stickers. Maybe for your information, the Bishop is the patron for the 660 hospital. This is an Anglican founded hospital so that’s why the bishop had to come in.” (Line 658-660).

Response: The Table has been edited to incorporate the recommendations by the reviewer. The identifiable traits have been deleted. Please see the updated Table 2 of the edited manuscript.

MINOR

1. Can the authors describe the duration of lockdowns in Lines 90-95?

Response: The duration of the lockdowns have been described in detail in the edited manuscript. It now appears from line 90-107 as “This was after recommendations of self-quarantine declared from 10th March 2020 for all travellers for two weeks were not feasible to containing the COVID-19 threat in the country. The government closed all the Ugandan borders on 23rd March 2020. With sprouting COVID-19 cases, the authorities suspended all public transport on 25th March 2020. This was later followed by a nationwide lockdown and night curfews for the first time in Uganda for two weeks from 1st April 2020. Before this, there had only been regional lockdowns like in the early 2000s to contain the Ebola outbreaks and civil wars seen around 1980 to 1985. All outdoor exercises were banned on 8th April 2020. After the two weeks, the Ugandan authorities extended the lockdown on the 14th April 2020 up to 5th May 2020. Though eased a bit with reduction on the travel restrictions, the lockdown was extended for another two weeks. The lockdown was finally eased on 4th June 2020 but the curfew measures were left in place to date. The COVID-19 pandemic took the country by surprise[6, 7]”.

2. Please clarify in Line 88 is these are only COVID-19 tested and confirmed cases? Are there estimates of suspected case load that could be described?

Response: Thanks for the concern; these are only COVID-19 tested and confirmed cases. We couldn’t access the total number of suspected cases as they are not reported fully 

3. Is there documentation of impact on MNCH attendance?

Response: Thanks for the concern. Documentation of the impact on MNCH attendance is in another paper with quantitative data. 

4. In the methods, then authors note this was part of a “bigger” study? Please clarify in the methods that this was an embedded qualitative study. Also, can the authors describe how they selected the 25 participants?

Response: “Embedded qualitative study has been added in the revised manuscript. It now appears in line 141 as “We conducted this embedded qualitative study as part of a bigger study that assessed the impact of COVID-19 pandemic on the provision of Maternal and Newborn healthcare services in eight health facilities in Kampala, Uganda between June 2020 and December 2020”.

5. Was any software used for data analysis?

Response: Due to financial constraints, we were unable to use any software for data analysis.

Reviewer #2: 1. The manuscript is probably technically sound but clarifications are needed in the methods to confirm this statement.

3. The authors state that they have made all the data available but it is obvious that they have not nor could they do this. This is a textual data set consisting of direct transcripts of many hours of interviewing. Making it available would violate the subjects' confidentiality and privacy. Best to state this in the manuscript that saying that all the data have become available.

Response: Thanks for the advice. We shall edit the write up as recommended by the reviewer.

4. The results and discussion sections are presented in an intelligible fashion but the intro and methods are not. The reviewer suggests how to improve in the comments that follow.

Response: Thanks for the honest observation. We have improved the write up in the introduction and methods section.

Additional comments by the reviewer are provided below:

1. The strength of the manuscript lies it its reported data.

2. Please, increase the readability of the data in two ways:

a. Improve the quality of writing in the intro and methods so that the reader has stamina left to make it to the results. This reviewer suggests ways to do that in more detail further down in their review.

b. Improve the readability of the results section by:

i. Creating a table of contents of the themes and subthemes that emerged from the data; state in one sentence a summary of what was found in each theme or subtheme; and the page number where the theme or subtheme is described. You currently have over 13 different themes, with subthemes under some of the themes – it is impossible for the reader to follow. Please number all themes on this table and in the narrative.

Response: Thanks for the recommendation. Table 3 has been added in the revised manuscript highlighting the themes, subthemes and the page numbers. Some of the redundant themes and subthemes have also been removed to shorten the manuscript as recommended by the reviewer.

ii. Going over each theme and thinking of ways to shorten either the text or the length of the direct quote to convey only what is said in a unique way that the text has not said. Current length is over 23 pages – bring it down to 12 pages.

Response: Thanks for the recommendation. Some of the redundant themes and subthemes have also been removed to shorten the manuscript as recommended by the reviewer.

3. Safeguard the anonymity and confidentiality of the data by removing the names of Hospitals and the age of each discussant from Table 1. For the purposes of the publication, the categories that you mention of “two Private hospitals, three Private-Not-for Profit hospitals and three Public health facilities” are sufficient for the reader to understand the study context. Replace hospital name by Private hospital 1, Private hospital 2, Private not for Profit hospital 1, Private not for Profit hospital 2, and so on. You report the mean age and range in the text so no need to specify on the Table. Instead of nurse 1, nurse 2, report the number of nurses, admins, and other providers interviewed in each hospital on the Table. By doing these changes you also increase the readability of current Table 1.

Response: Thanks for the advice. The names of the hospitals, age of the discussants have been removed from Table 1.

4. Methods

a. Describe all the teams that were involved in the study in one place. It looks like you had a data collection team, a data analysis team, and a team of external or independent (unclear) investigators that was checking the rigor. Like this: Team 1 had this and that investigator and did this; Team 2.. and so on.

Response: Clarity has been added in the revised manuscript. The changes now appear in line 253-65 of the revised manuscript as “We had three teams on the study. Team 1 was in charge of data collection. The team was composed of two researchers and two field note takers. The two researchers had doctoral degrees and were familiar with the local hospital settings. This team had research training for three days. They were trained on how to identify and interview potential clients. They were also trained on participant recruitment while observing the research ethics in accordance to the Declaration of Helsinki [26]. The two field note takers were fluent in English and Luganda, the locally spoken language. Team 2 was in charge of data analysis. It was composed of Principal investigator and one administrator. This team had to ensure transcription accuracy and data analysis. Team 3 was composed of two independent researchers whose task was to read and review the content (interview transcripts, and field notes word-for-word, line-for-line) several times for quality checks and triangulation”.

b. For rigor, you examined the credibility, confirmability, transferability and dependability of the data; also their congruency, triangulation and trustworthiness. These are all noble goals but your description of each is so confusing to any reader who might like to replicate your work at their hospitals, to the point of giving up reading the manuscript right there. You may clarify that these are related but slightly different aspects of data validity and you made sure to check for all of them by having a separate team constantly comparing the field notes, the transcripts, the codes, and their interpretation. Your manuscript does not describe any other evidence to compare your data with, neither do you ever mention that you took videos which brings on another layer of ethics and IRB approval issues. Please clean up and shorten that section.

Response: Thanks for the advice, manuscript has been edited accordingly and the video aspects edited out as we didn’t have ethical approval for it. However clarity has been added on the fact that different teams did the comparisons of the field notes, the transcripts, the codes and their interpretations in line 355-7. It now appears as “Field notes and transcripts, codes and their interpretations were made by separate teams of investigators”. 

5. I will now further elaborate the changes needed in the intro that I mention under comment 2.a.

1. Ref 1 – you need a HWO reference for this

Response: The reference has been edited accordingly

2. Ref 2 – need full citation (page numbers, issue)

Response: The reference has been edited accordingly

3. Ref 3 need the complete citation, weblink, and when it was accessed

4. Ref 4 , 5 – same

5. “with many vulnerable 90 populations, 1.4 million HIV positive patients, 800,000 diabetic patients and 100,000 TB sputum 91 patients, the President of Uganda, Yoweri Kaguta with his cabinet initially closed public 92 gatherings, shopping malls, and public domestic and international travels from the 18th March 93 2020.”

a. The first part of this sentence belongs in the next paragraph on how the health care system in Uganda was already overwhelmed, and needs the citation of ref 9. The second part of this sentence doesn’t need the name of the President for a scientific publication but rather a reference from a government website.

Response: The paragraph has been adjusted as advised by the reviewer and the President’s name removed as recommended. It now appears as “With 1.4 million HIV positive patients, 800,000 diabetic patients and 100,000 TB positive sputum patients, the Ugandan health system was already overstretched[11].

b. Use a better term for TB sputum patients

Response: “TB sputum positive patients” has been adopted in the revised manuscript in line 93.

6. Ref 6 and 7. The reader needs to know the full citation. Is this a scientific journal? A newspaper? Provide the weblink and when it was accessed.

7. Ref 8 should be a ref on Uganda prior to the pandemic – this a WHO reference on the pandemic. Best to only refer to 9 there (line 100). Ref 8 is appropriate in line 107.

Response: The references have been adjusted as recommended.

8. “The Ministry of Health in Uganda has already reported a decline in the current immunization coverage 104 though the overall impact of 105 COVID-19 on the immunization coverage is yet to be determined.” Need a citation or rephrase.

Response: The sentence has been improved to “The Ministry of Health in Uganda has already reported a decline in the current immunization coverage during the COVID-19 pandemic[16]” in line 115-7 in the revised manuscript.

9. Ref 6 and 12 are the same. Streamline.

Response: Thanks for the observation; we have edited out reference 12.

10. Ref 11. Need to know the type of document, weblink and date accessed.

11. Ref 13. The reader needs to know if this is a book, a journal, a report, the weblink and when it was accessed.

Response: More clarity has been added to these references.

12. “In Uganda, 4,600 women deliver everyday” rephrase – perhaps you mean to say that there are xx deliveries per day.

Response: The sentence has been changed to “In Uganda, there are 4,600 deliveries per day [17-19] in line 119 of the revised manuscript.

13. “Interruption in access to quality maternal 108 and newborn health services could put over 10,000 lives of these women and their babies in 109 danger.” Show your calculations and define the time period.

Response: The sentence has been changed in line 121-3 in the revised manuscript to “Interruption in access to quality maternal and newborn health services with the travel restrictions in place to curb the COVID 19 could put over 10,000 lives of both women and their babies in danger every single day of the COVID 19 pandemic”. This takes into account the multiple gestation that stretches the number above 10,000 for the mothers and their babies

14. “Due to shortages in the personal protective equipment in a number of health facilities, many 112 health workers offering maternal and newborn services fear for their lives.” This sentence already gives your study results away. It belongs to the results section. Delete from intro. Write it in the past tense in the results.

Response: The sentence has been deleted as recommended by the reviewer. 

15. “some health workers can’t access their work stations easily 118 with the COVID-19 travel restrictions” This gives away the data – put in the results section. Use past tense in your writing throughout to avoid being outdated and to maintain tense sequence throughout the manuscript.

Response: The sentence has been deleted from the introduction.

16. Format of ref 15 and 16 is good. Make all refs read like this.

17. “Some reports reveal health workers having been assaulted 119 by security personnel as they try to access or leave their work stations especially during curfew 120 hours [12, 15, 16]. “ Define security personnel. Do you mean police, hospital security, private firms? But this sentence also gives away your results – best to delete and let your data speak. Again, use past tense.

Response: The sentence has been deleted as advised by the reviewer.

18. Ref 15 is about women in labour not health workers. Use an appropriate reference.

Response: The sentence has been deleted as it’s about women in labour and not healthcare providers.

19. “some health workers have further have been deployed to manage the COVID-19 patients [9].” You cannot use a 2016 reference to substantiate a statement about 2020 or 2021. Also fix the grammar.

Response: The reference has been changed. The grammar has also been fixed. It now appears in the revised manuscript in line 134-6 as “Despite the low human resource available for maternal and newborn health, some health workers were deployed to manage the COVID-19 patients[12]

20. “with the aim of streamlining patient care in similar future disasters.” This is a good aim, but you also need an aim on streamlining patient care during the current disaster which still lingers in a big way. Edit abstract, intro and discussion on this.

Response: The sentence has been changed to “It’s against this background that we sought to understand the lived experiences and perceptions of the health workers offering maternal and newborn services during the COVID-19 pandemic in Uganda with the aim of streamlining patient care in the current and similar future disasters”.

6. Here I elaborate the changes needed in the methods under comment 2.a.

1. Reference the larger study, even if only manuscript in preparation

Response: The reference of the bigger study has been added as advised by the reviewer.

2. “phenomenological and inductive thematic approaches” Need a ref for phenomenological and for inductive.

Response: The references have been added as recommended by the reviewer. It now appears in the revised manuscript as “We used the phenomenological [24]and inductive thematic approaches[25] to explore the lived experiences and perspectives of healthcare providers as they offer maternal and newborn services in the eight selected facilities in Kampala”.

3. ““Quality was determined through retrospective 136 review of hospital records on maternal and newborn services that included; hospital deliveries, 137 antenatal attendances, immunization coverage, family planning services offered, postnatal clinic 138 attendance, HIV care services six months before and during the COVID-19 pandemic. Using 139 interviewer-administered questionnaires; we also collected patient quantitative data to assess any 140 trends in the care offered during the COVID-19 pandemic.” You do not need this statement. It refers to a different study. Delete.

Response: The sentence has been deleted as advised by the reviewer.

4. “We opted for in depth interviews 141 instead of focus group discussions to ensure social distancing and minimize spread of the 142 COVID-19.” There are certain strengths that focus groups have as a data collection method compared to in depth interviews, and vice versa. Say what you missed by not conducting focus groups due to the reasons that you cite and what you gained by conducting IDIs. The statement as is shows lack of understanding of focus groups as a qualitative research method.

5. “We conducted 50 in depth interviews. Twenty five among pregnant and 143 breastfeeding women to assess their lived experiences, perceptions about the quality of services 144 offered to them during the pandemic with the ultimate goal of identifying gaps and what their 145 level of satisfaction was as they sought care during the COVID-19 pandemic.“ This sentence does not belong to this manuscript. Here describe the methods you used only for this manuscript, i.e. the 25 interviews with health providers.

Response: The sentence has been deleted from the revised manuscript.

6. Ref 17 needs completion. I will now stop repeating this comment. Please check all refs and provide the weblink and when it was accessed.

7. Ref 18, 19, 22 and 24 are not needed.

8. “merging” you mean “emerging”

Response: The references have been deleted as recommended by the reviewer. 

8. “The 261 interviews were tape recorded and transcribed verbatim” check and eliminate repetitions. This phrase appears under data collection, under quality control, and under data analysis. The recording should be mentioned under data collection only. The transcription under data analysis only.

Response: The transcription phrase has been removed from the data collection and put under the data analysis. The recording has been mentioned only under the data collection as recommended by the reviewer. 

10. “To ensure trustworthiness and credibility, two independent researchers read and reviewed the content (interview transcripts, 290 and field notes word-for-word, line-for291 line) several times. Field notes and interview transcripts from each of the interviews were 292 assessed individually and later integrated to strengthen the data analysis and dependability of the 293 study findings. “This is stated under data analysis but under Data collection it is also stated that “All of the interviews were tape recorded and 255 transcribed verbatim immediately thereafter. Transcription accuracy was ensured at the end of 256 the interviews by the Principal investigator and one Administrator. Field notes and the 257 transcription were compared for congruency.” Best to eliminate from both data analysis and data collection sections and describe in simple words as I suggest in 4.b under Rigor only.

Response: These phrases have been deleted from both the data collection and analysis sections as recommended by the reviewer.

11. “We 307 thereafter identified themes reflecting on the depth. We later compared with other classes so as to 308 delimit the theories and achieve conceptual congruency[31].” Explain to the reader what type of depth you reflected on, and what were these other classes and theories that you mention. Better yet, delete.

Response: This phrase has been deleted as recommended by the reviewer.

12. “We observed data credibility by 312 ensuring checks by two members of the research team to accept codes from the transcription” This you did for confirmability. The credibility was already established because you had the recordings to prove it. Delete and summarize as I suggest in 4.b.

13. “peer debriefing from the senior researchers” unclear what this did – probably transferability on whether the data from one hospital was similar for another hospital? Clarify or delete and summarize as I suggest in 4.b.

Response: These phrases have been deleted as recommended by the reviewer.

14. You mention an “external investigator” in line 324 and independent investigators” in line 327. Were these investigators different from the PI, and admin? Best to delete whole section and follow my suggestion under 4.a.

Response: This phrase has been deleted as recommended by the reviewer.

15. Results: Lines 330 to 333 are repetitive - delete. Start section by saying that Table 1 describes the types of cadres interviewed… and edit Table 1 according to my suggestion 3.

16. Results: “The participants 342 generally had a number of similar experiences in regards to maternal and newborn health service343 delivery irrespective of the nature of health facility they worked in.” Since you took such great efforts on rigor, it should have been those multiple checkers who said that the data were comparable among all hospitals in the study and that the participants had similar experiences which allowed for their collective reporting. State that in the results, with the exception of your first theme on preference for working in Private, not for Profit or public facility, where the lived experiences differed.

Response: Line 330-3 and first theme have been deleted as recommended by the reviewer.

END OF COMMENTS

6. PLOS authors have the option to publish the peer review history of their article (what does this mean?). If published, this will include your full peer review and any attached files.

Do you want your identity to be public for this peer review? For information about this choice, including consent withdrawal, please see our Privacy Policy.

Reviewer #1: No

Reviewer #2: No

23rd June 2021:

Editor’s comments

Thank you for submitting your manuscript entitled "Lived Experiences of Frontline Healthcare Providers offering Maternal and Newborn Services amidst the Corona virus Disease 19 Pandemic in Uganda: A Qualitative study" to PLOS ONE. Your manuscript files have been checked in-house but before we can proceed we need you to address the following issues:

1) At this time, please confirm that your submission contains your "minimal data set", which PLOS defines as consisting of the data set used to reach the conclusions drawn in the manuscript with related metadata and methods, and any additional data required to replicate the reported study findings in their entirety. This includes:

i) The values behind the means, standard deviations and other measures reported;

ii) The values used to build graphs;

iii) The points extracted from images for analysis.

Response: Thanks for the comments. The minimal data set is included in the submitted manuscript. Our study is purely qualitative and extensive quantitative data is not included in the manuscript. There are no graphs or images for analysis in our write up. We have however added age to Table 2 and standard deviation to the mean of the participants in the descriptive statistics in the results section of the revised manuscript line 258.

Response: Supporting files are uploaded as supporting information however the data contains sensitive information in the dataset which can be traced back to the participants. It’s ethical if the information is only availed on request to protect the study participants.

2) Please amend the title either on the online submission form or in your manuscript so that they are identical.

Response: The title has been amended as recommended.

3) Please amend your list of authors on the manuscript to ensure that each author is linked to an affiliation.

We note that you have included affiliation numbers 1,2,3,4,5 and 6 however no affiliations have authors linked to them. 

Response: The affiliations have been amended as advised by the reviewer.

Your manuscript has been returned to your account. Please log on to PLOS Editorial Manager at https://www.editorialmanager.com/pone/ to access your manuscript.

Your manuscript can be found in the "Revisions Sent Back to the Author" link under the New Submissions menu. After you have made the changes requested above, please be sure to view and approve the revised PDF after rebuilding the PDF to complete the resubmission process.

Please note that these changes have been requested to comply with submission guidelines and your manuscript will *not* be sent to review until you have fully adhered to our requests. Once your paper has been seen by an Editor we may return it to you for further information or amendments.

 28th June 2021

TO: THE PLOS ONE EDITORIAL TEAM

Object: PONE-D-21-03659 Lived Experiences of Frontline Healthcare Providers offering Maternal and Newborn Services amidst the Corona virus Disease 19 Pandemic in Uganda: A Qualitative study

With great pleasure we are thankful for your consideration of our manuscript for publication in your reputable journal. In response to the editors’ comments sent to us on 28th June 2021, we have revised the above manuscript accordingly. 

Editor’s comments

Thank you for submitting your manuscript entitled "Lived Experiences of Front line Healthcare Providers offering Maternal and Newborn Services amidst the Corona virus Disease 19 Pandemic in Uganda: A Qualitative study" to PLOS ONE. Your manuscript files have been checked in-house but before we can proceed we need you to address the following issues:

1) Please amend the title either on the online submission form or in your manuscript so that they are identical.

We note that in your manuscript your title reads: Lived Experiences of Frontline Healthcare Providers offering Maternal and Newborn Services amidst the Novel Corona virus Disease 19 Pandemic in Uganda: A Qualitative study

We note that on the online submission form you title reads: Lived Experiences of Front line Healthcare Providers offering Maternal and Newborn Services amidst the Corona virus Disease 19 Pandemic in Uganda: A Qualitative study.

Response: Thanks for the comments. The title in the online submission has been edited to “Lived Experiences of Frontline Healthcare Providers offering Maternal and Newborn Services amidst the Novel Corona virus Disease 19 Pandemic in Uganda: A Qualitative study”

---

## [Decision Letter · Decision Letter 1]

5 Aug 2021

PONE-D-21-03659R1

Lived Experiences of Frontline Healthcare Providers offering Maternal and Newborn Services amidst the Novel Corona virus Disease 19 Pandemic in Uganda: A Qualitative study

PLOS ONE

Dear Dr. Kayiga,

Thank you for submitting your manuscript to PLOS ONE. After careful consideration, we feel that it has merit but does not fully meet PLOS ONE’s publication criteria as it currently stands. Therefore, we invite you to submit a revised version of the manuscript that addresses the points raised during the review process.

Thank you for this revision. Unfortunately there are still outstanding comments from the previous submission, as well as new comments from this revision. As suggested by Reviewer #2, please utilize a three column table and list out each reviewer comment in column one, your response in column two, and the page and line numbers where the change was made in column three.

We look forward to receiving your revised manuscript.

Kind regards,

Michelle L. Munro-Kramer, PhD, CNM, FNP-BC

Academic Editor

PLOS ONE

Journal Requirements:

Additional Editor Comments (if provided):

I suggest paying careful attention to all three reviewers' comments, but would specifically like to highlight:

1) The need to use consistency in language (e.g., COVID, COVID-19, Corona Virus). After defining the Coronavirus disease (in the first paragraph of the introduction), please select one term and use consistently in the abstract and text.

2) The first two references are about the United States. Please ensure you are using appropriate global resources (ideally from the WHO) when describing the global pandemic.

3) It is not a journal requirement to include data (especially if potentially identifiable). I agree with Reviewer #2 that there are a number of factors that compromise the confidentiality of the sample (e.g., the raw data file, listing the hospital/health facility names and locations, specificity about participant characteristics per site). I would recommend removing Table 1 and summarizing characteristics of the type of facilities included (but not needing to name them). I would summarize Table 2 to list characteristics overall (e.g., number of males, females; range and mean for age, etc.) based on hospital type as recommended by Reviewer #4. Consider removing the individual names of hospitals and thanking the administrators more generally. Finally, it is not necessary to submit ethical approval forms, interview questions, and the raw data. I would consider removing these.

4) Please note the suggestions to the rigor section described by Reviewer #2.

5) Table 3 is an excellent addition to the manuscript.

We look forward to receiving your revised submission.

Reviewers' comments:

Reviewer's Responses to Questions

**Comments to the Author**

1. If the authors have adequately addressed your comments raised in a previous round of review and you feel that this manuscript is now acceptable for publication, you may indicate that here to bypass the “Comments to the Author” section, enter your conflict of interest statement in the “Confidential to Editor” section, and submit your "Accept" recommendation.

Reviewer #2: (No Response)

Reviewer #3: (No Response)

Reviewer #4: (No Response)

2. Is the manuscript technically sound, and do the data support the conclusions?

Reviewer #2: Partly

Reviewer #3: Yes

Reviewer #4: Yes

3. Has the statistical analysis been performed appropriately and rigorously? 

Reviewer #2: N/A

Reviewer #3: N/A

Reviewer #4: N/A

4. Have the authors made all data underlying the findings in their manuscript fully available?

Reviewer #2: Yes

Reviewer #3: Yes

Reviewer #4: No

5. Is the manuscript presented in an intelligible fashion and written in standard English?

Reviewer #2: Yes

Reviewer #3: Yes

Reviewer #4: Yes

6. Review Comments to the Author

Reviewer #2: 1. Many improvements evident throughout, most notably the addition of Table 3

2. Key words: suggest including "lived experiences"

3. Publication Ethics Table 1: by mentioning the names and identifying information of each study hospital continues to reveal who the study participants were and compromises their confidentiality. Please delete hospital names and identifying location information (highway or hill name, founding details, division name; does it matter in this case whether it was Anglican or Catholic?). Keep in minimal information that helps the reader understand what type of hospital with what type of capabilities and catchment was studied.

4. Publication Ethics: Same is true of the supporting information linked to the manuscript. Other than the participant interview guide, the documents contain hospital identifying information and therefore inappropriate to share with the journal’s readership. This includes:

a. Clearance forms

b. Protocol

c. Consent form health workers

d. Raw data set – every quote in the data set shows which hospital it came from; if you have to share, then all data should be de-identified, like you did in quotes used in the results section. This is a lot of work – I would rather you did not share due to data confidentiality issues.

e. The other forms are not needed to be made public either and should be omitted from the supporting information.

5. The TASO IRB approval letter and the rstug UNCSTRefNumber do not contain identifying information but they are not needed by the readership either and should also be omitted as unnecessary.

6. “All participants were sanitized” A better way to say this for people is that they practiced a disinfection protocol prior to the interview and hopefully their interviewers did that as well.

7. “Team 3 was composed of two independent 161 researchers whose task was to read and review the content (interview transcripts, and field 162 notes word-for-word, line-for-line) several times for quality checks and triangulation.” Do you mean to review transcripts and notes compared to the recordings for transcription accuracy? Team 2 was already doing that. I suggest you rephrase the description of Team 3 to say that this team was checking rigor according to the Lincoln – Guba criteria and leave it at that because you explain those further down. What you say about triangulation fits better under rigor – move it there and say how team 3 was doing triangulation of the analysis findings comparing data from one hospital with another hospital (because I haven’t seen any other method used for triangulation).

8. “The research materials were kept under restricted access by only authorized staff for 182 patient confidentiality and privacy” There were no patients in the study. Please clarify that it was about the confidentiality of the participants, now violated by Table 1 and supporting documents.

9. “Participants were reimbursed for participating in the study in form of transport refunds and 190 refreshments.” Perhaps clarify that the refreshments were to take home? They were all in masks and PPE.

10. “The interviews were transcribed verbatim immediately. Transcription accuracy was ensured 198 at the end of the interviews by the Principal investigator and one Administrator. Field notes 199 and the transcription were compared for congruency. Data collection and analysis were 200 conducted concurrently until data saturation was achieved. This was done so that insights 201 from the data analysis could be used to make the required adjustments in the interview guide 202 and evaluate the credibility of the emerging themes in the subsequent interviews.” This was already stated under quality control. Delete the repetitions and merge the rest with that section.

11. “Data was 203 coded and analyzed manually using a framework matrix developed using an Excel workbook.” This statement implies that you developed the codes deductively and put them into a pre-designed framework. It is in contrast with the detailed description right below on how you developed inductive codes according to phenomenology. Did you perhaps built the framework matrix in excel after that detailed and careful process? State that and move that sentence after the description of the code development, to line 215.

12. “We ensured long term 220 involvement of the research team with the healthcare providers.” What rigor criterion does this fulfill? I suggest delete unless you use it to say that you are confident that you established good rapport with the participants.

The authors did not follow my previous feedback on how to painlessly summarize rigor. I therefore provide feedback below on how to improve the accuracy and readability of the rigor section:

13. “Data dependability was 221 ensured by having team 3 that was devoted to continuous reading through of the transcripts to 222 ensure ongoing comparison of the key information generated during the data collection and 223 analysis processes.” This is about ensuring that the findings of the analysis were aligned with the data collected in the transcripts and is what you do for the credibility criterion. Please edit.

14. “Dependability was observed by the stringent coding procedure and inter 224 coder corroboration.” Here please add that you made sure to document what each code meant in detail as illustrated in Table 3, so that another researcher could replicate this coding under a similar context (dependability criterion).

15. “Data transferability was observed by ensuring that participants’225 statements were captured with barely any modifications made yet ensuring a rich, thick 226 description of the study process by the research team.” The first part of this sentence is about confirmability by capturing the statements of the participants without any modifications and use of quotes as you do in the results section. The second part of the sentence about the thick description of the study context is what you provide in the intro and tables 1 and 2 so the findings could be transferable by the reader to another context if it were similar to the one of the hospitals in Kampala (transferability criterion). Please edit so it is clear to the reader.

16. “Thorough checks of procedures and 227 results were emphasized to improve the dependability and transferability of the data [19].” This is good but if you clarify the sentences above it you don’t need to repeat.

17. “Confirmability was observed by comparing the results to other evidence and field notes by an 229 external investigator [34] who read and compared the study results with the field notes and 230 memos.” Comparing the results with other evidence is triangulation. Either say what other evidence you were comparing the results with or delete. The rest about comparing the results with field notes and memos is confirmability - merge with the other part on confirmability. But this is the first time you mention memos. Say under data analysis how you were also writing memos and for what purpose, or delete memos from the sentence.

18. “We ensured that the coordinators of the interviews or discussions didn’t participate in 231 the analysis but critiqued the results from the analysis and ensured that these results 232 conformed to their expectations from the discussions.” It is not a criterion of rigor that field researchers stay away from the analysis. What you did is “member checking”, i.e. another way of validating the findings. Please state.

19. “Field notes and transcripts, codes and 233 their interpretations were made by separate teams of investigators.” Again this is not a criterion of rigor. You can delete. Or merge under data analysis.

20. “More than 90% of the healthcare 239 providers…” You should not use percentages when the denominator is so small (n=25). Simply say the great majority of participants…

21. Publication Ethics Table 2: too much information that compromises the confidentiality of the participants and doesn’t improve the dependability and transferability of the results. Please look at my suggestions in the previous rounds on how to improve Table 2. You can collectively report average age and say that all had bachelors degrees and up. Why is marital status relevant here? Delete.

22. “The participants 242 generally had a number of similar experiences in regards to maternal and newborn health 243 service delivery irrespective of the nature of health facility they worked (Table 3).” Here is where you can say that your data comparisons during rigor analysis showed a number of similar experiences irrespective of the health facility.

23. Table 3 is very good and helps a lot. “Much stigma was associated with contracting COVID-19. This meant no working for more than 2 weeks for the infected healthcare providers.” This statement needs clarification. How could infected providers work for 2 weeks?

24. “A special773 vote of thanks goes to the Administrators of the eight health facilities namely; Kawempe774 National Referral hospital, Kawaala Health Centre III, China Uganda Friendship Hospital,775 Naguru (Naguru Hospital), St. Francis Hospital Nsambya, Lubaga Hospital, Mengo Hospital, 776 Kampala Hospital, and Case Hospital for all the support they gave us during the study period.” Here again you are compromising the confidentiality of your participants without improving dependability or transferability of the results. You can instead anonymously thank all the participating hospitals.

25. References 1 and 2 continue to be inappropriate. You can’t use studies from the US to support a statement of WHO. You need a WHO reference for that.

26. References 6, 7, 8, 10, 11, 14, 15, 37, 39, 40, 44, 49, 51 need a web link and date accessed

27. Reference 8 and 13 are the same reference. Eliminate one of the 2 and provide weblink and date accessed.

28. References 21-29 are not needed when you delete the hospital names.

29. References 38, 41, 50 missing volume issue page info.

30. A recommendation on how to best address the reviewers’ comments without missing any, is to create a two-column table where on the one side you list each comment and on the right side you insert your response and direct quote from the manuscript.

END OF COMMENTS

Reviewer #3: Thank you for submitting this revised paper. You have obviously undertaken extensive revissions.

My comments are only minor. These include:

a) Introduction:

i) Good to benchmark the extensive lockdown and other measures in Uganda at the start of the pandemic to those seen in other Africa countries (Ogunleye OO et al. Response to the Novel Corona Virus Pandemic Across Africa: Successes, Challenges, and Implications for the Future. Frontiers in pharmacology. 2020;11:1205) helping to reduce mortality - certainly when compared to e.g. a number of Western European countries

ii) Good to include more up-to-date figures for COVID-19 than late January. In addition % in WHO Africa vs. rest of the world (this builds on i)

iii) Lines 84 - 85 - I assume you mean 'Uganda' by 'U'. In addition - I do believe Uganda was more prepared than a number of other countries including e.g. US

iv) Line 91 - A similar situation on reduced routine vaccinations across Africa - please see Abbas K et al. Routine childhood immunisation during the COVID-19 pandemic in Africa: a benefit-risk analysis of health benefits versus excess risk of SARS-CoV-2 infection. The Lancet Global health. 2020;8(10):e1264-e72

v) Line 104 - avoid unscientific terms such as 'grossly' throughout the paper - better to say 'appreciably' than 'grossly'

b) Discussion - I would concentrate on the key areas as well as say what the authorities in Uganda should now do as a result of your findings for this and future pandemic. This does not come through clearly enough. This does not mean adding to the Discussion - merely making it more focused. The same applies to the Conclusion. This would enhance the utility of the paper in Uganda, across Africa and across LMICs

Reviewer #4: It is a pleasure to review the study entitled : “Lived Experiences of Frontline Healthcare Providers offering Maternal and Newborn Services amidst the Novel Corona virus Disease 19 Pandemic in Uganda: A Qualitative study”. This paper’s strength is in the richness of the data and the in-depth descriptions of the challenges faced by maternal and newborn healthcare providers in Uganda during the pandemic, and how that influenced care provision. The study is also a platform to raise healthcare providers’ voices about the horrible experiences and negative treatment that they received, and to share their opinions of recommendations to continue care provision during the pandemic and beyond. It is also difficult not to appreciate the rigorous research methodology that was applied. Despite the witnessed improvements in the structure of the manuscript after the first revision, there remained some issue that can be addressed before publication. My two main comments are:

- Although I do believe that the rich and expressive quotes are a strength of the manuscript, they do tend to make the results’ section a lot longer than it can be. Some of them are particularly long and repetitive of the text summarizing the results and therefore can be either shortened or deleted altogether (for example the one in line 300 – 304 can be deleted). Perhaps keeping one quote per theme is sufficient, and the reader can always refer to the “raw data” supporting information for more.

- The manuscript describes the lived experiences during the COVID-19 pandemic. Yet, as we all have witnessed, the pandemic has been ongoing for almost 16 months, and with varying levels of restrictions over time. In the manuscript, the time frame of the described “lived experience” is not clear: was it the early phase of the pandemic (first lockdown), or does it stretch to include the period of data collection? The authors can be more specific about the recall period. This can be very relevant especially considering the second lockdown that Uganda is recently going through, to see whether any of the lessons learned from the first lockdown have helped in managing the second response.

Other minor comments are noted below, divided by section:

Supporting information:

- File called “Raw data” : suggestion to change the name of the file to : ”Detailed summary of the data by theme with quotes” since it is not possible to share the raw data (i.e. complete transcripts of interviews) due to issues of privacy and anonymity. Naming the file “Raw data” gives the false impression that the dull transcripts are actually published.

- File called “Participant interview guide” : it seems that this file contains questions addressed to women who have sought care and not to healthcare providers, and the questions do not match those mentioned in the response to the reviewers. Suggestion to please revise and align.

Abstract

- Suggest to rephrase this sentence: “With the travel restrictions, social distancing associated with the containment of the virus, the maternal and newborn healthcare service in Uganda could be inaccessible, unaffordable, and unavailable to both the healthcare providers and many pregnant or laboring women.” It seems like the care is unaffordable and unavailable to healthcare providers - Is this intentional? – consider using the space in the abstract to focus mainly on the barriers faced by healthcare providers

- This is a qualitative study and usually do not use terms such as “primary outcome” (which has more of a quantitative connotation). It is already clear in the objectives what the “outcome” of the study is. Suggestion to rephrase as: “the interview guide primarily explored xxxx”

- The first sentence in the conclusion is probably correct but it is not a direct observation of this research. The conclusion can focus more on healthcare providers’ wellbeing and ability to provide care, and the need to respect and support them rather than about the service delivery

Background:

- Line 84-85: The COVID-19 pandemic took U by surprise[8, 10].

o Although I agree, it did take “me” and everyone by surprise, but I think the authors mean “took Uganda”

o Suggest to move this sentence to the beginning of the paragraph

- Line 93-96: “Interruption in access to quality maternal and newborn health services with the travel restrictions in place to curb the COVID-19, could put over 10,000 lives of both women and their babies in danger every single day of the COVID-19 pandemic.”

o Please provide a reference to this estimate

o This paragraph could use a bit more information about the MNH situation in Uganda before the pandemic: e.g. maternal mortality rate, skilled birth attendance, facility birth coverage, ANC coverage etc. how did these aspects evolve over time? And why is COVID-19 a particular threat to them, especially if it’s affecting healthcare providers.

o The background is also missing information about the structure of the health system in Uganda before the pandemic – where do women usually seek care (hospitals, healthcare centres?) how is care covered? Public vs private sector role in the health system? And how are they similar/different to each other? etc.

- Line 98: there is a “15” misplaced after December 2020. Also not clear what this sentence adds: “if it means that healthcare workers were infected with COVID-19”? Please clarify , with more details about the number of healthcare workers if that is possible.

Methods:

- Line 121: is this a public health facility? Not a hospital?

- Table 1:

o Suggest to present similar and complete information on all the hospitals ; e.g. why is number of deliveries per year available for Kawaala health centre and not others? If possible recommend to add for all

o Suggestion to divide the “description” column into more structured columns, for example: level of care (primary, secondary, tertiary) ; some proxy of size of the health facility (e.g. number of maternity beds or number of deliveries in the past year – depending on which info is readily available); number of maternal and newborn healthcare providers (total or estimate); operating hours; free vs. paid services

o An important characteristic to mention about the hospitals is whether or not they treated any pregnant women / women in labour who were suspected/confirmed with COVID-19

- Explain a bit more about the selection of the hospitals (purposive sampling of the biggest hospitals in Kampala, from three sectors public, private, private not-for-profit)

- Line 121 – 123: “These eight facilities were the biggest service providers for public and private maternal and newborn health care in Kampala.” – why past tense here “were”? Suggest to change to present

- Line 139: “All participants were sanitized”. Suggest to rephrase to : participants sanitized their hands or strict hand washing and sanitization were required from all participants…

- It is important to mention where the interviews took place in the methods section: was it at the health facilities where participants worked? Or at the researchers office?

- Line 155: “potential clients”. Suggest to rephrase to potential participants

- Line 168: do the authors mean: no new emerging themes?

- Quality control: what happened with the data from the pilot interviews? Was it included in the analysis? Please be clear about that and if yes why? If no why not?

- Line 180: data were backed-up? Where and how?

Results:

- Title of the heading: please remove “baseline”

- Table 2 is not completely showing on the page (please format and resize)

- Please align the use of “obstetrician/gynecologist” vs. “medical doctor” when describing the cadres in the methods, results and table 2

- Suggestion for table 2: to switch the rows and the columns (the three types of facilities become columns, so that the categories of each variable are not repeated every time)

Private Public Private not for profit

Sex

Male 2 2 2

female 4 8 7

Age

20-29 0 1 1

30-39 4 5 4

>40 2 4 4

- It is useful to know how many interviews were done per facility – perhaps could be added to table 1?

- Table 3 – it is not clear for the reader why the page number is added to the table. Also keeping in mind that this might change when the paper is published, I suggest to remove this column. Or it can be used to indicate a reference to the “raw data” supplementary material if necessary

- The authors indicate that HCP experienced a number of similar themes across the facilities, but did they note any discrepancies or similarities within the health facilities? E.g. differences between cadres who work at the same hospital? (just our of curiosity about dynamics between different cadres)

- The perception that patient numbers increased is interesting- despite the fact that we could have assumed the opposite to happen (blocked roads/fear of healthcare seeking in facilities).

- Comment/suggestion: try to avoid “quantitative” terms in the results (e.g. change “a significant number” on line 254 to something like “many/most” etc.

- Line 258: exclamation mark after “gloves”. Suggest to remove to the keep the results description as objective as possible.

- Suggestion to always refer to it as “COVID-19”. Sometimes COVID alone is used (it is ok if it’s in a quote, but not the main text). Line 451: suspected to have COVID-19

- Suggestion to spell out CME in-text (first occurrence line 529)

Discussion

- Line 650-651: specify which services exactly

- Paragraph lines 685-697: recommendation about telemedicine should be considered with caution as it can lead to inequality in accessibility (poverty, illiteracy among women) and its impact on the quality of maternity care is not yet well understood.

Suggest to revise the list of abbreviations and align with the updated version of the manuscript as some terms were deleted e.g. PMTCT

7. PLOS authors have the option to publish the peer review history of their article (what does this mean?). If published, this will include your full peer review and any attached files.

Reviewer #2: No

Reviewer #3: **Yes: **Brian Godman

Reviewer #4: No

---

## [Author Response · Author response to Decision Letter 1]

6 Sep 2021

AUTHORS’ RESPONSE TO REVIEWS

TITLE: Lived Experiences of Frontline Healthcare Providers offering Maternal and Newborn Services amidst the Novel Corona virus Disease 19 Pandemic in Uganda: A Qualitative study

Authors

Herbert Kayiga (hkayiga@gmail.com)

Diane Achanda Genevive (achandadiane@gmail.com)

Pauline Mary Amuge (paulacallista@gmail.com)

Andrew Sentoogo Ssemata (andrewssemata@yahoo.co.uk)

Racheal Samantha Nanzira (nanzirasamanthar@gmail.com)

Annettee Nakimuli (Annettee.nakimuli@gmail.com)

Version: 4

Date: 9th August 2021

Authors’ response to reviews: See over

 1st September 2021

TO: THE PLOS ONE EDITORIAL TEAM

Object: PONE-D-21-03659 Lived Experiences of Frontline Healthcare Providers offering Maternal and Newborn Services amidst the Corona virus Disease 19 Pandemic in Uganda: A Qualitative study

With great pleasure we are thankful for your consideration of our manuscript for publication in your reputable journal. In response to the editors’ comments sent to us on 5th August 2021, we have revised the above manuscript accordingly. 

Editor’s comments

Journal Requirements:

Response: Thanks for the comment. The reference list has been revised accordingly.

Additional Editor Comments (if provided):

I suggest paying careful attention to all three reviewers' comments, but would specifically like to highlight:

1) The need to use consistency in language (e.g., COVID, COVID-19, Corona Virus). After defining the Coronavirus disease (in the first paragraph of the introduction), please select one term and use consistently in the abstract and text.

Response: Thanks for the recommendation; COVID-19 is now the consistent term used in the revised version of the manuscript.

2) The first two references are about the United States. Please ensure you are using appropriate global resources (ideally from the WHO) when describing the global pandemic.

Response: The first two references (Garg, S., et al., Hospitalization rates and characteristics of patients hospitalized with laboratory-confirmed coronavirus disease 2019—COVID-NET, 14 States, March 1–30, 2020. Morbidity and mortality weekly report, 2020. 69(15): p. 458.

2. Team, C.C.-R., et al., Geographic differences in COVID-19 cases, deaths, and incidence—United States, February 12–April 7, 2020. Morbidity and Mortality Weekly Report, 2020. 69(15): p. 465-471) have been replaced with new references as published by the World Health Organization (Sohrabi, C., et al., World Health Organization declares global emergency: A review of the 2019 novel coronavirus (COVID-19). International journal of surgery, 2020. 76: p. 71-76.

2. Purcell, L.N. and A.G. Charles, An Invited Commentary on “World Health Organization declares global emergency: A review of the 2019 novel Coronavirus (COVID-19)": Emergency or new reality? International journal of surgery (London, England), 2020. 76: p. 111).

It is not a journal requirement to include data (especially if potentially identifiable). I agree with Reviewer #2 that there are a number of factors that compromise the confidentiality of the sample (e.g., the raw data file, listing the hospital/health facility names and locations, specificity about participant characteristics per site). I would recommend removing Table 1 and summarizing characteristics of the type of facilities included (but not needing to name them). 

Response: Thanks so much for the comment. Table 1 from the original draft was removed and revised as recommended in the Rebuttal letter of 7th June 2021. The current table 2 doesn’t reveal the identity of the participants.

I would summarize Table 2 to list characteristics overall (e.g., number of males, females; range and mean for age, etc.) based on hospital type as recommended by Reviewer #4. Consider removing the individual names of hospitals and thanking the administrators more generally. Finally, it is not necessary to submit ethical approval forms, interview questions, and the raw data. I would consider removing these.

Response: Thanks for the recommendation. The ethical approval forms, interview questions and the raw data will be removed.

4) Please note the suggestions to the rigor section described by Reviewer #2.

Response: The suggestions have been noted.

5) Table 3 is an excellent addition to the manuscript.

Response: Thanks for the observation.

We look forward to receiving your revised submission.

Reviewers' comments:

Reviewer's Responses to Questions

Comments to the Author:

1. If the authors have adequately addressed your comments raised in a previous round of review and you feel that this manuscript is now acceptable for publication, you may indicate that here to bypass the “Comments to the Author” section, enter your conflict of interest statement in the “Confidential to Editor” section, and submit your "Accept" recommendation.

Reviewer #2: (No Response)

Reviewer #3: (No Response)

Reviewer #4: (No Response)

2. Is the manuscript technically sound, and do the data support the conclusions?

Reviewer #2: Partly

Reviewer #3: Yes

Reviewer #4: Yes

3. Has the statistical analysis been performed appropriately and rigorously?

Reviewer #2: N/A

Reviewer #3: N/A

Reviewer #4: N/A

4. Have the authors made all data underlying the findings in their manuscript fully available?

Reviewer #2: Yes

Reviewer #3: Yes

Reviewer #4: No

5. Is the manuscript presented in an intelligible fashion and written in standard English?

Reviewer #2: Yes

Reviewer #3: Yes

Reviewer #4: Yes

6. Review Comments to the Author:

Reviewer Comment Response to Comment Page No. and Line 

Reviewer 2#

Many improvements evident throughout, most notably the addition of Table 3

2. Key words: suggest including "lived experiences"

3. Publication Ethics Table 1: by mentioning the names and identifying information of each study hospital continues to reveal who the study participants were and compromises their confidentiality. Please delete hospital names and identifying location information (highway or hill name, founding details, division name; does it matter in this case whether it was Anglican or Catholic?). Keep in minimal information that helps the reader understand what type of hospital with what type of capabilities and catchment was studied.

4. Publication Ethics: Same is true of the supporting information linked to the manuscript. Other than the participant interview guide, the documents contain hospital identifying information and therefore inappropriate to share with the journal’s readership. This includes:

a. Clearance forms

b. Protocol

c. Consent form health workers

d. Raw data set – every quote in the data set shows which hospital it came from; if you have to share, then all data should be de-identified, like you did in quotes used in the results section. This is a lot of work – I would rather you did not share due to data confidentiality issues.

e. The other forms are not needed to be made public either and should be omitted from the supporting information.

5. The TASO IRB approval letter and the study UNCST Ref Number do not contain identifying information but they are not needed by the readership either and should also be omitted as unnecessary.

6. “All participants were sanitized” A better way to say this for people is that they practiced a disinfection protocol prior to the interview and hopefully their interviewers did that as well.

7. “Team 3 was composed of two independent 161 researchers whose task was to read and review the content (interview transcripts, and field 162 notes word-for-word, line-for-line) several times for quality checks and triangulation.” Do you mean to review transcripts and notes compared to the recordings for transcription accuracy? Team 2 was already doing that. I suggest you rephrase the description of Team 3 to say and leave it at that because you explain those further down. What you say about triangulation fits better under rigor – move it there and say how team 3 was doing triangulation of the analysis findings comparing data from one hospital with another hospital (because I haven’t seen any other method used for triangulation).

8. “The research materials were kept under restricted access by only authorized staff for 182 patient confidentiality and privacy” There were no patients in the study. Please clarify that it was about the confidentiality of the participants, now violated by Table 1 and supporting documents.

9. “Participants were reimbursed for participating in the study in form of transport refunds and 190 refreshments.” Perhaps clarify that the refreshments were to take home? They were all in masks and PPE.

10. “The interviews were transcribed verbatim immediately. Transcription accuracy was ensured 198 at the end of the interviews by the Principal investigator and one Administrator. Field notes 199 and the transcription were compared for congruency. Data collection and analysis were 200 conducted concurrently until data saturation was achieved. This was done so that insights 201 from the data analysis could be used to make the required adjustments in the interview guide 202 and evaluate the credibility of the emerging themes in the subsequent interviews.” This was already stated under quality control. Delete the repetitions and merge the rest with that section.

11. “Data was 203 coded and analyzed manually using a framework matrix developed using an Excel workbook.” This statement implies that you developed the codes deductively and put them into a pre-designed framework. It is in contrast with the detailed description right below on how you developed inductive codes according to phenomenology. Did you perhaps built the framework matrix in excel after that detailed and careful process? State that and move that sentence after the description of the code development, to line 215.

12. “We ensured long term 220 involvement of the research team with the healthcare providers.” What rigor criterion does this fulfill? I suggest delete unless you use it to say that you are confident that you established good rapport with the participants.

The authors did not follow my previous feedback on how to painlessly summarize rigor. I therefore provide feedback below on how to improve the accuracy and readability of the rigor section:

13. “Data dependability was 221 ensured by having team 3 that was devoted to continuous reading through of the transcripts to 222 ensure ongoing comparison of the key information generated during the data collection and 223 analysis processes.” This is about ensuring that the findings of the analysis were aligned with the data collected in the transcripts and is what you do for the credibility criterion. Please edit.

14. “Dependability was observed by the stringent coding procedure and inter 224 coder corroboration.” Here please add that you made sure to document what each code meant in detail as illustrated in Table 3, so that another researcher could replicate this coding under a similar context (dependability criterion).

15. “Data transferability was observed by ensuring that participants’225 statements were captured with barely any modifications made yet ensuring a rich, thick 226 description of the study process by the research team.” The first part of this sentence is about confirmability by capturing the statements of the participants without any modifications and use of quotes as you do in the results section. The second part of the sentence about the thick description of the study context is what you provide in the intro and tables 1 and 2 so the findings could be transferable by the reader to another context if it were similar to the one of the hospitals in Kampala (transferability criterion). Please edit so it is clear to the reader.

16. “Thorough checks of procedures and 227 results were emphasized to improve the dependability and transferability of the data [19].” This is good but if you clarify the sentences above it you don’t need to repeat.

17. “Confirmability was observed by comparing the results to other evidence and field notes by an 229 external investigator [34] who read and compared the study results with the field notes and 230 memos.” Comparing the results with other evidence is triangulation. Either say what other evidence you were comparing the results with or delete. The rest about comparing the results with field notes and memos is confirmability - merge with the other part on confirmability. But this is the first time you mention memos. Say under data analysis how you were also writing memos and for what purpose, or delete memos from the sentence.

18. “We ensured that the coordinators of the interviews or discussions didn’t participate in 231 the analysis but critiqued the results from the analysis and ensured that these results 232 conformed to their expectations from the discussions.” It is not a criterion of rigor that field researchers stay away from the analysis. What you did is “member checking”, i.e. another way of validating the findings. Please state.

19. “Field notes and transcripts, codes and 233 their interpretations were made by separate teams of investigators.” Again this is not a criterion of rigor. You can delete. Or merge under data analysis.

20. “More than 90% of the healthcare 239 providers…” You should not use percentages when the denominator is so small (n=25). Simply say the great majority of participants…

21. Publication Ethics Table 2: too much information that compromises the confidentiality of the participants and doesn’t improve the dependability and transferability of the results. Please look at my suggestions in the previous rounds on how to improve Table 2. You can collectively report average age and say that all had bachelors degrees and up. Why is marital status relevant here? Delete.

22. “The participants 242 generally had a number of similar experiences in regards to maternal and newborn health 243 service delivery irrespective of the nature of health facility they worked (Table 3).” Here is where you can say that your data comparisons during rigor analysis showed a number of similar experiences irrespective of the health facility.

23. Table 3 is very good and helps a lot. “Much stigma was associated with contracting COVID-19. This meant no working for more than 2 weeks for the infected healthcare providers.” This statement needs clarification. How could infected providers work for 2 weeks?

24. “A special773 vote of thanks goes to the Administrators of the eight health facilities namely; Kawempe774 National Referral hospital, Kawaala Health Centre III, China Uganda Friendship Hospital,775 Naguru (Naguru Hospital), St. Francis Hospital Nsambya, Lubaga Hospital, Mengo Hospital, 776 Kampala Hospital, and Case Hospital for all the support they gave us during the study period.” Here again you are compromising the confidentiality of your participants without improving dependability or transferability of the results. You can instead anonymously thank all the participating hospitals.

25. References 1 and 2 continue to be inappropriate. You can’t use studies from the US to support a statement of WHO. You need a WHO reference for that.

26. References 6, 7, 8, 10, 11, 14, 15, 37, 39, 40, 44, 49, 51 need a web link and date accessed

27. Reference 8 and 13 are the same reference. Eliminate one of the 2 and provide weblink and date accessed.

28. References 21-29 are not needed when you delete the hospital names.

29. References 38, 41, 50 missing volume issue page info.

30. A recommendation on how to best address the reviewers’ comments without missing any, is to create a two-column table where on the one side you list each comment and on the right side you insert your response and direct quote from the manuscript.

Thanks for the kind observation.

“Lived experiences” has been added to the key words

The hospital names and identifying location information has been deleted from the revised manuscript as recommended.

Documents containing hospital identifying information, clearance forms, protocol, consent forms and raw data sets will not be shared as recommended.

The UNCST, TASO IRB forms will also be retrieved as recommended by the reviewer.

The manuscript has been edited accordingly. It now appears as “Disinfection protocols were observed prior to the interviews.

The write up has been edited accordingly as advised by the reviewer. It now appears as “Team 3 was composed of two independent researchers whose task was checking rigor according to the Lincoln – Guba criteria”.

The manuscript has been edited accordingly as advised by the reviewer. It now appears as “Triangulation was checked by team 3 that was devoted to continuous reading through of the transcripts to ensure ongoing comparison of the key information generated from one hospital to another during the data collection and analysis processes”.

Correction has been made in the revised manuscript. The hospital details and supporting information has been edited as recommended. This now appears in the revised manuscript as “The research materials were kept under restricted access by only authorized staff for participant confidentiality and privacy”

Thanks for the keen observation. Since the participants were paid for both, to avoid the confusion, in the write up, refreshment has been deleted. It now appears as “Participants were reimbursed for participating in the study in form of transport refunds”.

The portion has been deleted as recommended by the reviewer.

Thanks for the recommendation. The manuscript has been edited accordingly. It now appears as “Data was coded and analyzed manually using a framework matrix developed using an Excel workbook built after a detailed and careful process of the merging codes’.

The sentence has been deleted as recommended by the reviewer.

Thanks again for the kind comment. The manuscript has been edited accordingly. It now appears as “Triangulation was checked by team 3 that was devoted to continuous reading through of the transcripts to ensure ongoing comparison of the key information generated from one hospital to another during the data collection and analysis processes”.

Thanks for the advice; this manuscript has been revised to “We made sure to document what each code meant in detail as illustrated in Table 3”.

As advised by the reviewer, transferability has been changed to confirmability as the sentence captures this aspect as opposed to transferability.

The revised manuscript now appears as “Data confirmability was observed by ensuring that participants’ statements were captured with barely any modifications made. Data transferability was ensured by the research team so that a rich, thick description of the study process was documented to enable replicability in a similar context elsewhere. 

Revision has been made in the sentences above as advised by the reviewer. To avoid repetition, the sentence has been deleted from the revised manuscript.

The aspect about confirmability has been deleted as advised

This sentence has been removed from the section on “Rigor” and shifted to ‘Quality control” since it’s not relevant under rigor.

It now appears as “We ensured that the coordinators of the interviews or discussions didn’t participate in the analysis but critiqued the results from the analysis and ensured that these results conformed to their expectations from the discussions. This was done to validate the study findings and also ensure quality in the study”. 

The sentence deleted as advised.

Thanks for the caution; the sentence has been revised accordingly to “The great majority of healthcare providers had more than ten years’ experience offering maternal and newborn health services”.

Table 2 has been improved as recommended by the reviewer. Marital statuses, level of education and age have been deleted accordingly. Age has been reported based on the average age and level of education as advised.

The sentence has been improved as advised by the reviewer. It now appears as “Data comparisons during rigor analysis showed a number of similar experiences in maternal and newborn health service delivery irrespective of the health facility (Table 3)”.

Clarity has been added to the write up in the revised manuscript. It now appears as “Stigma was associated with contracting COVID-19. Infected healthcare providers were side-lined for two or more weeks. This meant no earning during the period of self-isolation or quarantine”.

Thanks for the keen observation. The manuscript has been edited accordingly. This aspect now appears as “A special vote of thanks goes to the Administrators of the health facilities that participated in the study”.

Thanks for the advice. WHO references have been used for reference 1 and 2.

1. Sohrabi, C., et al., World Health Organization declares global emergency: A review of the 2019 novel coronavirus (COVID-19). International journal of surgery, 2020. 76: p. 71-76.

2. Purcell, L.N. and A.G. Charles, An Invited Commentary on “World Health Organization declares global emergency: A review of the 2019 novel Coronavirus (COVID-19)": Emergency or new reality? International journal of surgery (London, England), 2020. 76: p. 111.

Thanks so much for the advice. A web link and date accessed have been added in the revised manuscript

References 21-29 have been deleted from the revised manuscript

The volume and page info has been added as advised.

A 2-column table has been created as recommended by the reviewer to address the raised comments and their responses. 

Page 3, Line 59

Table 1: Page 8, line 169,170 

Page 15, Line 182-183

Page 16, Line 202-203

Page 18, Line 263-266

Page 17, Line 229-231

Page 17, Line 238

Page 17, Line 218-231

Page 18, Line 255-257

Page 18, Line 263-267

Page 18-19, Line 267-268

Page 19, Line 268-271

Page 17, Line 224-227

Page 19, Line 278-279

Page 19, Line 280

Page 19-20, Line 281-283

Page 20, Line 288-290

Page 41, Line 754-755

Page 42, Line 789-795

Page 42-45, Line 789-963

Page 44-45

Reviewer #3: 

Reviewer Comment Response to Comment Page No. and Line 

Reviewer #3

Thank you for submitting this revised paper. You have obviously undertaken extensive revisions. My comments are only minor. These include:

a) Introduction:

i) Good to benchmark the extensive lockdown and other measures in Uganda at the start of the pandemic to those seen in other Africa countries (Ogunleye OO et al. Response to the Novel Corona Virus Pandemic Across Africa: Successes, Challenges, and Implications for the Future. Frontiers in pharmacology. 2020;11:1205) helping to reduce mortality - certainly when compared to e.g. a number of Western European countries

ii) Good to include more up-to-date figures for COVID-19 than late January. In addition % in WHO Africa vs. rest of the world (this builds on i)

iii) Lines 84 - 85 - I assume you mean 'Uganda' by 'U'. In addition - I do believe Uganda was more prepared than a number of other countries including e.g. US

iv) Line 91 - A similar situation on reduced routine vaccinations across Africa - please see Abbas K et al. Routine childhood immunization during the COVID-19 pandemic in Africa: a benefit-risk analysis of health benefits versus excess risk of SARS-CoV-2 infection. The Lancet Global health. 2020;8(10):e1264-e72

v) Line 104 - avoid unscientific terms such as 'grossly' throughout the paper - better to say 'appreciably' than 'grossly'

b) Discussion - I would concentrate on the key areas as well as say what the authorities in Uganda should now do as a result of your findings for this and future pandemic. This does not come through clearly enough. This does not mean adding to the Discussion - merely making it more focused. The same applies to the Conclusion. This would enhance the utility of the paper in Uganda, across Africa and across LMICs

Thanks for your kind remarks on the progress of the manuscript.

The manuscript has been revised carefully to highlight the measures that were undertaken in other low and middle income settings at the start of the pandemic. This now appears in the revised manuscript as “With lessons from other low and middle income settings, like Vietnam where lockdowns, extensive contact tracing and social distance had resulted in barely any mortality attributed to COVID-19[7, 8], Uganda instigated a nationwide lockdown to contain the COVID-19 pandemic. 

With no clear cure to COVID-19 [9, 10], like other African countries like South Africa, Malawi, South Sudan, Kenya, Ghana, Nigeria, and Rwanda[11], Uganda took a number of measures to contain the COVID-19 pandemic.

Thanks for the caution. Up-to-date figures have been used in the revised manuscript. It now appears in the revised manuscript as “As of 10th August 2021, globally there were 202.1 million confirmed cases of COVID-19 and 4.29 million deaths [3]. In Africa as of 10th August 2021, there were 5.14 million confirmed cases and 122,025 deaths from COVID-19, which is lower than the 78.6 million confirmed cases and 2.03 million deaths in Americans [3] and 61.2 million confirmed cases in Europe with 1.23 million deaths from COVID-19[3].

Uganda reported her first COVID-19 case on the 21st March 2020[4]. Since then the number of confirmed cases had reached 95,723 as of 06th August 2021 with 2,783 deaths reported by the Uganda Ministry of Health [5].

‘U’ is now written in full in the revised manuscript. The manuscript has been edited accordingly to justify the implementation of the lockdown in Uganda. It now appears “With lessons from other low and middle income settings, like Vietnam where lockdowns, extensive contact tracing and social distance had resulted in barely any mortality attributed to COVID-19[7, 8], and more case fatality rates from COVID-19 in the United States [9] and Europe [10]where preventative measures were not fully implemented, Uganda instigated a nationwide lockdown to contain the COVID-19 pandemic.

The manuscript has been edited accordingly. The write up now appears as “COVID-19 has exerted enormous pressure on National Health Service programs in many African countries like Expanded Program on Immunization [17] as result of closure of some of the vaccination clinics with some of the healthcare providers put in quarantine when suspected or confirmed with COVID-19 or shifted to manage COVID-19 patients[13].

Despite evidence of routine childhood immunization benefit over COVID-19 associated risks with the vaccination clinics[18], the Ministry of Health of Uganda has already reported a decline in the current immunization coverage during the COVID-19 pandemic [19]. Similar trends in immunization coverage have also been reported in South Sudan, Zimbabwe, South Africa and Nigeria [13].

Thanks so much for the advice. ‘Grossly’ has been changed to ‘appreciably’ in the edited version of the manuscript.

The discussion and conclusion write up has been edited as advised by the reviewer. 

Page 5, Line 89-93

Page 4, Line 65-74

Page 4, Line 74,84-96

Page 6, Line 125-133

Page 7, Line 147

Page 36-40

Reviewer #4: 

Reviewer Comment Response to Comment Page No. and Line 

It is a pleasure to review the study entitled : “Lived Experiences of Frontline Healthcare Providers offering Maternal and Newborn Services amidst the Novel Corona virus Disease 19 Pandemic in Uganda: A Qualitative study”. This paper’s strength is in the richness of the data and the in-depth descriptions of the challenges faced by maternal and newborn healthcare providers in Uganda during the pandemic, and how that influenced care provision. The study is also a platform to raise healthcare providers’ voices about the horrible experiences and negative treatment that they received, and to share their opinions of recommendations to continue care provision during the pandemic and beyond. It is also difficult not to appreciate the rigorous research methodology that was applied. Despite the witnessed improvements in the structure of the manuscript after the first revision, there remained some issue that can be addressed before publication. 

My two main comments are:

- Although I do believe that the rich and expressive quotes are a strength of the manuscript, they do tend to make the results’ section a lot longer than it can be. Some of them are particularly long and repetitive of the text summarizing the results and therefore can be either shortened or deleted altogether (for example the one in line 300 – 304 can be deleted). Perhaps keeping one quote per theme is sufficient, and the reader can always refer to the “raw data” supporting information for more.

- The manuscript describes the lived experiences during the COVID-19 pandemic. Yet, as we all have witnessed, the pandemic has been ongoing for almost 16 months, and with varying levels of restrictions over time. In the manuscript, the time frame of the described “lived experience” is not clear: was it the early phase of the pandemic (first lockdown), or does it stretch to include the period of data collection? The authors can be more specific about the recall period. This can be very relevant especially considering the second lockdown that Uganda is recently going through, to see whether any of the lessons learned from the first lockdown have helped in managing the second response.

Other minor comments are noted below, divided by section:

Supporting information:

- File called “Raw data” : suggestion to change the name of the file to : ”Detailed summary of the data by theme with quotes” since it is not possible to share the raw data (i.e. complete transcripts of interviews) due to issues of privacy and anonymity. Naming the file “Raw data” gives the false impression that the dull transcripts are actually published.

- File called “Participant interview guide” : it seems that this file contains questions addressed to women who have sought care and not to healthcare providers, and the questions do not match those mentioned in the response to the reviewers. Suggestion to please revise and align.

Abstract

- Suggest to rephrase this sentence: “With the travel restrictions, social distancing associated with the containment of the virus, the maternal and newborn healthcare service in Uganda could be inaccessible, unaffordable, and unavailable to both the healthcare providers and many pregnant or laboring women.” It seems like the care is unaffordable and unavailable to healthcare providers - Is this intentional? – consider using the space in the abstract to focus mainly on the barriers faced by healthcare providers

- This is a qualitative study and usually do not use terms such as “primary outcome” (which has more of a quantitative connotation). It is already clear in the objectives what the “outcome” of the study is. Suggestion to rephrase as: “the interview guide primarily explored xxxx”

- The first sentence in the conclusion is probably correct but it is not a direct observation of this research. The conclusion can focus more on healthcare providers’ wellbeing and ability to provide care, and the need to respect and support them rather than about the service delivery

Background:

- Line 84-85: The COVID-19 pandemic took U by surprise[8, 10].

o Although I agree, it did take “me” and everyone by surprise, but I think the authors mean “took Uganda”

o Suggest to move this sentence to the beginning of the paragraph

- Line 93-96: “Interruption in access to quality maternal and newborn health services with the travel restrictions in place to curb the COVID-19, could put over 10,000 lives of both women and their babies in danger every single day of the COVID-19 pandemic.”

o Please provide a reference to this estimate

o This paragraph could use a bit more information about the MNH situation in Uganda before the pandemic: e.g. maternal mortality rate, skilled birth attendance, facility birth coverage, ANC coverage etc. how did these aspects evolve over time? And why is COVID-19 a particular threat to them, especially if it’s affecting healthcare providers.

o The background is also missing information about the structure of the health system in Uganda before the pandemic – where do women usually seek care (hospitals, healthcare centres?) how is care covered? Public vs. private sector role in the health system? And how are they similar/ different to each other? etc.

- Line 98: there is a “15” misplaced after December 2020. Also not clear what this sentence adds: “if it means that healthcare workers were infected with COVID-19”? Please clarify , with more details about the number of healthcare workers if that is possible.

Methods:

- Line 121: is this a public health facility? Not a hospital?

- Table 1:

o Suggest to present similar and complete information on all the hospitals ; e.g. why is number of deliveries per year available for Kawaala health centre and not others? If possible recommend to add for all

o Suggestion to divide the “description” column into more structured columns, for example: level of care (primary, secondary, tertiary) ; some proxy of size of the health facility (e.g. number of maternity beds or number of deliveries in the past year – depending on which info is readily available); number of maternal and newborn healthcare providers (total or estimate); operating hours; free vs. paid services

o An important characteristic to mention about the hospitals is whether or not they treated any pregnant women / women in labour who were suspected/confirmed with COVID-19

- Explain a bit more about the selection of the hospitals (purposive sampling of the biggest hospitals in Kampala, from three sectors public, private, private not-for-profit)

- Line 121 – 123: “These eight facilities were the biggest service providers for public and private maternal and newborn health care in Kampala.” – why past tense here “were”? Suggest to change to present

- Line 139: “All participants were sanitized”. Suggest to rephrase to : participants sanitized their hands or strict hand washing and sanitization were required from all participants…

- It is important to mention where the interviews took place in the methods section: was it at the health facilities where participants worked? Or at the researchers office?

- Line 155: “potential clients”. Suggest to rephrase to potential participants

- Line 168: do the authors mean: no new emerging themes?

- Quality control: what happened with the data from the pilot interviews? Was it included in the analysis? Please be clear about that and if yes why? If no why not?

- Line 180: data were backed-up? Where and how?

Results:

- Title of the heading: please remove “baseline”

- Table 2 is not completely showing on the page (please format and resize)

- Please align the use of “obstetrician/gynecologist” vs. “medical doctor” when describing the cadres in the methods, results and table 2

- Suggestion for table 2: to switch the rows and the columns (the three types of facilities become columns, so that the categories of each variable are not repeated every time)

Private Public Private not for profit

Sex

Male 2 2 2

female 4 8 7

Age

20-29 0 1 1

30-39 4 5 4

>40 2 4 4

- It is useful to know how many interviews were done per facility – perhaps could be added to table 1?

- Table 3 – it is not clear for the reader why the page number is added to the table. Also keeping in mind that this might change when the paper is published, I suggest removing this column. Or it can be used to indicate a reference to the “raw data” supplementary material if necessary

- The authors indicate that HCP experienced a number of similar themes across the facilities, but did they note any discrepancies or similarities within the health facilities? E.g. differences between cadres who work at the same hospital? (just our of curiosity about dynamics between different cadres)

- The perception that patient numbers increased is interesting- despite the fact that we could have assumed the opposite to happen (blocked roads/fear of healthcare seeking in facilities).

- Comment/suggestion: try to avoid “quantitative” terms in the results (e.g. change “a significant number” on line 254 to something like “many/most” etc.

- Line 258: exclamation mark after “gloves”. Suggest to remove to the keep the results description as objective as possible.

- Suggestion to always refer to it as “COVID-19”. Sometimes COVID alone is used (it is ok if it’s in a quote, but not the main text). Line 451: suspected to have COVID-19

- Suggestion to spell out CME in-text (first occurrence line 529)

Discussion

- Line 650-651: specify which services exactly

- Paragraph lines 685-697: recommendation about telemedicine should be considered with caution as it can lead to inequality in accessibility (poverty, illiteracy among women) and its impact on the quality of maternity care is not yet well understood.

Suggest to revise the list of abbreviations and align with the updated version of the manuscript as some terms were deleted e.g. PMTCT Thanks so much for your kind remarks.

Thanks for the recommendation. The result section has been shortened by reducing the quotes in the write up in the revised manuscript

Clarity has been added to the phase of the lockdown from the abstract up to the method’s section. In the abstract, this appears as “The study sought to understand the experiences and perceptions of healthcare providers at the frontline during the first phase of the lockdown as they offer maternal and newborn health care services in both public and private health facilities in Uganda with the aim of streamlining patient care in face of the current COVID-19 pandemic and in future disasters”.

In the introduction, further clarity on when the phase of the lockdown has been added. It now appears as “

It’s against this background that the study sought to understand the lived experiences and perceptions of the health workers offering maternal and newborn services during the first phase of the lockdown to contain the COVID-19 pandemic during the COVID-19 pandemic in Uganda with the aim of streamlining patient care in the current and similar future disasters”

Then in the method’s section, the write up has been changed to “We conducted this embedded qualitative study as part of a bigger study that assessed the impact of COVID-19 pandemic on the provision of Maternal and Newborn healthcare services in eight health facilities in Kampala, Uganda between June 2020 and December 2020[30] during the first phase of the lockdown”.

Thanks for the advice. The file name has been edited from “Raw data” to ”Detailed summary of the data by theme with quotes”

Thanks for the keen observation. The interview guide has been changed.

The sentence has been edited in the Abstract to “With the travel restrictions, social distancing associated with the containment of theCOVID-19 pandemic, healthcare providers could be faced with challenges of accessing their work stations, and risk burn out as they offer maternal and newborn services”.

Thanks for the advice. The write up has been changed to “The interview guide primarily explored the lived experiences of healthcare providers as they offer maternal and newborn healthcare services during the COVID-19 pandemic”.

The first sentence in the conclusion has been edited as advised by the reviewer to “The COVID-19 Pandemic has led to a decline in quality of maternal and newborn service delivery by the healthcare providers as evidenced by shorter consultation time and failure to keep appointments to attend to patients…”.

“U” has been written as “Uganda” in the edited manuscript. The sentence has also been shifted as recommended. The write up now appears as “Uganda reported her first COVID-19 case on the 21st March 2020[4]. Since then the number of confirmed cases had reached 95,723 as of 06th August 2021 with 2,783 deaths reported by the Uganda Ministry of Health [5]. The COVID-19 pandemic took Uganda by surprise [6, 7]”.

A reference has been added as recommended by the reviewer. It now appears as “Interruption in access to quality maternal and newborn health services with the travel restrictions in place to curb the COVID-19 could put over 10,000 lives of both women and their babies in danger every single day of the COVID-19 pandemic [6]”. 

The manuscript has been revised accordingly to add information about the MNH situation in Uganda before the pandemic. It now appears as “Prior to the COVID-19, Uganda through strategies like five year Health Sector Strategic Plans for the past two decades had reduced maternal mortality rates from 500 in 2000 to 375 deaths per 100,000 live births [22, 23]. The four visit antenatal attendance was at 59.9% from 33.1% in 2011 [24]. The unmet need for modern contraception had reduced to 26% from 30% in 2016[23, 25]. The fertility rate in Uganda stands at 4.3 currently from 5.3 in 2000 [22]. The postnatal care was still below optimal levels in Uganda at 54.3% [23]. The neonatal mortality was at 19.9 per 1000 live births before COVID-19 from 33 deaths per 1000 live births [22]. Neonatal tetanus protection had reached 85% as compared 52% in 2000 [26]. Significant progress had also been seen in the BCG immunization at 1 year with 88% while that of Haemophilus influenzae type B (Hib) and Diphtheria, Pertussis (whooping cough), and Tetanus (DPT) vaccine coverage was at 93% [23] before COVID-19 pandemic”.

The background has been edited accordingly. The structure of the health system in Uganda is written out as “Healthcare in Uganda is offered mainly by public (70%), private-Not for profit (20%) and private health facilities (10%) [9, 10]. Public health facilities are structured in the following categories; National and Regional referral hospitals, general hospitals, district hospitals/ Health centre IVs (offering care to a population of 100,000 both in and outpatient services and emergency surgeries), Health Centre III (serving a population of 20,000 at the sub county level offering mainly outpatient and maternity services), Health centre II (serving a population of 5,000 and being run by an enrolled midwife) and the Health centre I (linking the community to the health system and being run by the village health teams with or without formal training). Care in the public facilities is free [11]. Private-Not-for Profit health facilities are mainly faith based facilities that offer care at a subsidized cost. The private health facilities are run by individuals or institutions with no exact control on how care is billed [12].

The sentence was meant to mention that 15 healthcare providers had died of covid-19 as of 31st December 2020. The revised manuscript now has even more updated figures. It appears as “As of 7th July 2021, 37 Ugandan health workers had died of COVID-19[30]”.

As suggested by other reviewers, the identity of the health facilities has been concealed.

Table 1 has been edited accordingly as recommended by the reviewer. More details have been added.

At the time of data collection, none of the selected health facilities were designed to treat COVID-19 patients. It’s the reason we didn’t capture this information in regards to admission of women with COVID-19 in our study. The ministry of Health had designated 2 facilities to manage the COVID-19 patients.

Thanks for the advice. The manuscript has been edited accordingly as advised by the reviewer. It now appears as “This study was conducted in eight health facilities (two Private hospitals, three Private-Not-for Profit hospitals and three Public health facilities) in Kampala, Uganda. These eight facilities were purposively selected because they are the biggest service providers in the three sectors (public, private-not-for profit and private) offering maternal and newborn health care in Kampala”.

The manuscript has been edited accordingly. It now appears as “Disinfection protocols were observed prior to the interviews.

The manuscript has been revised accordingly to highlight the sites for the in depth interviews. This now appears as “All the in depth interviews were administered in English, the official language used in Uganda in quiet rooms at the different selected health facilities as recommended by the hospital administrators”.

“Potential clients” has been rephrased as “potential participants” as recommended by the reviewer.

Correction has been made in the revised manuscript. It now appears as “After ascertaining data saturation with no new emerging themes, we stopped the data collection [37, 38].

Clarity has been added in the edited manuscript. It now appears as “A pilot study was carried out with four healthcare providers to pretest and modify the interview guide. Data from the pilot study was also included in the analysis as the healthcare providers in the pilot were not included in the main study”.

The edited manuscript has more clarity added in. It’s now written as “Data was backed up on hard drives, online databases and two computers”.

“Baseline” has been removed from the title. It now appears as “Characteristics of participants”

Table 2 has been formatted and resized. “Obstetrician/Gynecologist” has been used instead of “medical doctor” as recommended by the reviewer. 

Table 2 has been edited as recommended by the review.

Since the interviews are implied in the table 2 by the number of participants at each facility, for clarity in the result’s section, we have added the text “. Of the 25 interviews, six were in private; ten were in public, while nine were in private not-for profit health facilities”.

Thanks for the advice, though in the prior reviewer’s comments, this was the suggestion. I have deleted the column on the page numbers as this might change with publication. I’ll keep reference however for the raw data as advised.

Clarity has been added in the revised manuscript to address this concern. It now appears as “The following themes and subthemes were generated from the data analysis. Data comparisons during rigor analysis showed a number of similar experiences in maternal and newborn health service delivery irrespective of the health facility. There were however some disparities in the experiences within the different cadres. Nurses tended to use more of the public means when compared to the obstetricians/gynaecologists and administrators. The way the nurses, navigated through the hassle of transport to the workstations were different from the obstetricians/gynaecologists (Table 3).

Thanks for the keen observation; the manuscript has been edited to imply that the patient load was high for the available workforce as some of the healthcare providers couldn’t access their work stations during the pandemic. The sub heading has been changed to “High Patient turnover per the available workforce and Burnout”

“Significant” has been deleted from the revised manuscript as advised by the reviewer” 

The exclamation mark has been removed as advised by the reviewer.

Thanks for the observation; “COVID-19” has been adopted throughout the manuscript.

CME has been spelt out as advised by the reviewer.

The implied services have been added in the revised manuscript. It now appears as “As highlighted by the discussants, there’s a need for the government to mediate a friendly working environment between the security personnel and the medical fraternity to minimize on delays seen during the pandemic that could have grossly affected maternal and newborn service delivery like immunization, health facility deliveries, antenatal and postnatal clinic attendance [11-13, 68]”.

Thanks so much for advice but since it was recommendation from the healthcare providers, it would be respectful to have their views represented in the publication. More reading around this area will however be ensured.

The abbreviation list has been updated as advised by the reviewer. 

Page 20-34

Page 1 Line 28-32

Page 6 Line 149-152

Page 7 Line 155-158

Page 1 Line 25-28

Page 2 Line 35-37

Page 2 Line 52-60

Page 4 Line 74-76

Page 6-7 Line 137-141

Page 6 Line 114-125

Page 4-5 Line 74-90

Page 7 Line 143

Page 8-14, Line 171,172

Page 8, Line 165-169

Page 13: Line 184-185

Page 13-14: Line 185-187

Page 14: Line 200

Page 14-15, Line 211-212

Page 15, Line 222-224

Page 15, Line 231

Page 17, Line 275

Page 18

Page 18

Page 17, Line 277-279

Page 18-22, Line 290-292

Page 18, Line 283-289

Page 19

Page 22, Line 299, Page 23, line 326, Page 24, Line 362

Page 22, Line 303

Page 30, Line 518

Page 35, Line 647-652

Page 38-39, Line 739-752

---

## [Editor Report · Decision Letter 2]

28 Oct 2021

Lived Experiences of Frontline Healthcare Providers offering Maternal and Newborn Services amidst the Novel Corona virus Disease 19 Pandemic in Uganda: A Qualitative study

PONE-D-21-03659R2

Dear Dr. Kayiga,

We’re pleased to inform you that your manuscript has been judged scientifically suitable for publication and will be formally accepted for publication once it meets all outstanding technical requirements.

Kind regards,

Michelle L. Munro-Kramer, PhD, CNM, FNP-BC

Academic Editor

PLOS ONE

Additional Editor Comments (optional):

Thank you for your careful attention to all reviewer comments. The manuscript has been substantially improved and I am happy to accept it for publication.

---

## [Editor Report · Acceptance letter]

3 Dec 2021

PONE-D-21-03659R2 

Lived Experiences of Frontline Healthcare Providers offering Maternal and Newborn Services amidst the Novel Corona virus Disease 19 Pandemic in Uganda: A Qualitative study 

Dear Dr. Kayiga:

I'm pleased to inform you that your manuscript has been deemed suitable for publication in PLOS ONE. Congratulations! Your manuscript is now with our production department. 

Kind regards, 

on behalf of

Dr. Michelle L. Munro-Kramer 

Academic Editor

PLOS ONE